# Efficient Bayesian network structure learning via local Markov boundary search

**Ming Gao**
The University of Chicago
minggao@uchicago.edu

**Bryon Aragam**
The University of Chicago
bryon@chicagobooth

## Abstract

We analyze the complexity of learning directed acyclic graphical models from observational data in general settings without specific distributional assumptions. Our approach is information-theoretic and uses a local Markov boundary search procedure in order to recursively construct ancestral sets in the underlying graphical model. Perhaps surprisingly, we show that for certain graph ensembles, a simple forward greedy search algorithm (i.e. without a backward pruning phase) suffices to learn the Markov boundary of each node. This substantially improves the sample complexity, which we show is at most polynomial in the number of nodes. This is then applied to learn the entire graph under a novel identifiability condition that generalizes existing conditions from the literature. As a matter of independent interest, we establish finite-sample guarantees for the problem of recovering Markov boundaries from data. Moreover, we apply our results to the special case of polytrees, for which the assumptions simplify, and provide explicit conditions under which polytrees are identifiable and learnable in polynomial time. We further illustrate the performance of the algorithm, which is easy to implement, in a simulation study. Our approach is general, works for discrete or continuous distributions without distributional assumptions, and as such sheds light on the minimal assumptions required to efficiently learn the structure of directed graphical models from data.

## 1 Introduction

Learning the structure of a distribution in the form of a graphical model is a classical problem in statistical machine learning, whose roots date back to early problems in structural equations and covariance selection [12, 45, 46]. Graphical models such as Markov networks and Bayesian networks impose structure in the form of an undirected graph (UG, in the case of Markov networks) or a directed acyclic graph (DAG, in the case of Bayesian networks). This structure is useful for variety of tasks ranging from querying and sampling to inference of conditional independence and causal relationships, depending on the type of graph used. In practice, of course, this structure is rarely known and we must rely on *structure learning* to first infer the graphical structure. The most basic version of this problems asks, given $n$ samples from some distribution $P$ that is represented by a graphical model $G = (V, E)$, whether or not it is possible to reconstruct $G$.

In this paper, we study the structure learning problem for Bayesian networks (BNs). Our main contribution is a fine-grained analysis of a polynomial time and sample complexity algorithm for learning the structure of BNs with potentially unbounded maximum in-degree and without faithfulness. In particular, in our analysis we attempt to expose the underlying probabilistic assumptions that are important for these algorithms to work, drawing connections with existing work on local search algorithms and the conditional independence properties of $P$.

35th Conference on Neural Information Processing Systems (NeurIPS 2021), virtual.

## 1.1 Contributions

One of the goals of the current work is to better understand the minimal assumptions needed to identify and learn the structure of a DAG from data. Although this is a well-studied problem, existing theoretical work (see Section 1.2) relies on assumptions that, as we show, are not really necessary. In particular, our results emphasize generic probabilistic structure (conditional independence, Markov boundaries, positivity, etc.) as opposed to parametric or algebraic structure (linearity, additivity, etc.), and hence provide a more concrete understanding of the subtle necessary conditions for the success of this approach.

With this goal in mind, we study two fundamental aspects of the structure learning problem: Identifiability and Markov boundary search. On the one hand, we provide a weaker condition for identifiability compared to previous work, and on the other, we exhibit families of DAGs for which forward greedy search suffices to provably recover the parental sets of each node. More specifically, our contributions can be divided into several parts:

1. *Identifiability* (Theorem 3.1). We prove a new identifiability result on DAG learning. Roughly speaking, this condition requires that the entropy conditioned on an ancestral set $H(X_k \,|\, A)$ of each node in $G$ is dominated by one of its ancestors. An appealing feature of this assumption is that it applies to general distributions without parametric or structural assumptions, and generalizes existing ones based on second moments to a condition on the local entropies in the model. We also discuss in depth various relaxations of this and other conditions (Appendix C).

2. *Local Markov boundary search* (Algorithm 2, Proposition 4.1). We prove finite-sample guarantees for a Markov boundary learning algorithm that is closely related to the incremental association Markov blanket (IAMB) algorithm, proposed in Tsamardinos et al. [42]. These results also shed light on the assumptions needed to successfully learn Markov boundaries in general settings; in particular, we do not require faithfulness, which is often assumed.

3. *Structure learning* (Algorithm 1, Theorem 5.1). We propose an algorithm which runs in $O(d^3 r \log d)$ time and $O(d^2 r \log^3 d)$ sample complexity, to learn an identifiable DAG $G$ from samples. Here, $d$ is the dimension and $r \leq d$ is the depth of the DAG $G$, defined in Section 2.

4. *Learning polytrees* (Theorem 5.2). As an additional application of independent interest, we apply our results to the problem of learning polytrees [10, 37].

5. *Generalizations and extensions* (Appendix C). In the supplement, we have included an extensive discussion of our assumptions with many examples and generalizations to illustrate the main ideas. For example, this appendix includes relaxations of the positivity assumption on $P$, the main identifiability condition (Condition 1), the PPS condition (Condition 2), and extensions to general, non-binary distributions. We also discuss examples of the conditions and a comparison to the commonly assumed faithfulness condition.

Finally, despite a long history of related work on Markov blanket learning algorithms [e.g. 1, 34, 40], to the best of our knowledge there has been limited theoretical work on the finite-sample properties of IAMB and related algorithms. It is our hope that the present work will serve to partially fill in this gap.

## 1.2 Related work

Early approaches to structure learning assumed faithfulness (for the definition, see Appendix B; this concept is not needed in the sequel), which allows one to learn the Markov equivalence class of $P$ [7, 15, 23, 30, 39]. Under the same assumption and assuming additional access to a black-box query oracle, Barik and Honorio [3] develop an algorithm for learning discrete BNs. Barik and Honorio [4] develop an algorithm for learning the undirected skeleton of $G$ without assuming faithfulness. On the theoretical side, the asymptotic sample complexity of learning a faithful BN has also been studied [14, 51]. Brenner and Sontag [5] propose the SparsityBoost score and prove a polynomial sample complexity result, although the associated algorithm relies on solving a difficult integer linear program. Chickering and Meek [8] study structure learning without faithfulness, although this paper does not establish finite-sample guarantees. Zheng et al. [49, 50] transform the score-based DAG learning problem into continuous optimization, but do not provide any guarantees. Aragam et al. [2]

analyze the sample complexity of score-based estimation in Gaussian models, although this estimator is based on a nonconvex optimization problem that is difficult to solve.

An alternative line of work, more in spirit with the current work, shows that $G$ itself can be identified without assuming faithfulness [24, 36, 38, 48], although these methods lack finite-sample guarantees. Recently, Ghoshal and Honorio [19] translated the equal variance property of Peters and Bühlmann [36] into a polynomial-time algorithm for linear Gaussian models with polynomial sample complexity. Around the same time, Park and Raskutti [33] developed an efficient algorithm for learning a special family of distributions with *quadratic variance functions*. To the best of our knowledge, these algorithms were the first provably polynomial-time algorithm for learning DAGs that did not assume faithfulness. See also Ordyniak and Szeider [32] for an excellent discussion of the complexity of BN learning. The algorithm of Ghoshal and Honorio [19] has since been extended in many ways [6, 17, 20, 44].

Our approach is quite distinct from these approaches, although as we show, our identifiability result subsumes and generalizes existing work on equal variances. Additionally, we replace second-moment assumptions with entropic assumptions, which are weaker and have convenient interpretations in terms of conditional independence, which is natural in the setting of graphical models. Since many of the analytical tools for analyzing moments and linear models are lost in the transition to discrete models, our work relies on fundamentally different (namely, information-theoretic) tools in the analysis. Our approach also has the advantage of highlighting the important role of several fundamental assumptions that are somewhat obscured by the linear case, which makes strong use of the covariance structure induced by the linear model. Finally, we note that although information-theoretic ideas have long been used to study graphical models [e.g. 25–28], these works do not propose efficient algorithms with finite-sample guarantees, which is the main focus of our work.

## 2 Preliminaries

**Notation** We use $[d] = \{1, \ldots, d\}$ to denote an index set. As is standard in the literature on graphical models, we identify the vertex set of a graph $G = (V, E)$ with a random vector $X = (X_1, \ldots, X_d)$, and in the sequel we will frequently abuse notation by identifying $V = X = [d]$. For any subset $S \subset V$, $G[S]$ is the subgraph defined by $S$. Given a DAG $G = (V, E)$ and a node $X_k \in V$, $\mathrm{pa}(k) = \{X_j : (j, k) \in E\}$ is the set of parents, $\mathrm{de}(k)$ is the set of descendants, $\mathrm{nd}(k) := V \setminus \mathrm{de}(k)$ is the set of nondescendants, and $\mathrm{an}(k)$ is the set of ancestors. Analogous notions are defined for subsets of nodes in the obvious way. A source node is any node $X_k$ such that $\mathrm{an}(k) = \emptyset$ and a sink node is any node $X_k$ such that $\mathrm{de}(k) = \emptyset$. Every DAG admits a unique decomposition into $r$ layers, defined recursively as follows: $L_1$ is the set of all source nodes of $G$ and $L_j$ is the set of all sources nodes of $G[V \setminus \cup_{t=0}^{j-1} L_t]$. By convention, we let $L_0 = \emptyset$ and layer width $d_j = |L_j|$, the largest width $\max_j d_j = w$. An ancestral set is any subset $A \subset V$ such that $\mathrm{an}(A) \subset A$. The layers of $G$ define canonical ancestral sets by $A_j = \cup_{t=0}^{j} L_t$. Finally, the Markov boundary of a node $X_k$ relative to a subset $S \subset V$ is the smallest subset $m \subset S$ such that $X_k \perp\!\!\!\perp (S \setminus m) \mid m$, and is denoted by $\mathrm{MB}(X_k; S)$ or $m_{Sk}$ for short.

The entropy of a discrete random variable $Z$ is given by $H(Z) = -\sum_z P(Z = z) \log P(Z = z)$, the conditional entropy given $Y$ is $H(Z \mid Y) = -\sum_{z,y} P(Z = z, Y = y) \log P(Z = z \mid Y = y)$, and the mutual information between $Z$ and $Y$ is $I(Z; Y) = H(Z) - H(Z \mid Y)$. For more background on information theory, see Cover and Thomas [9].

**Graphical models** Let $X = (X_1, \ldots, X_d)$ be a random vector with distribution $P$. In the sequel, for simplicity, we assume that $X \in \{0, 1\}^d$ and that $P$ is strictly positive, i.e. $P(X = x) > 0$ for all $x \in \{0, 1\}^d$. These assumptions are not necessary; see Appendix C for extensions to categorical random variables and/or continuous random variables and nonpositive distributions.

A DAG $G = (V, E)$ is a *Bayesian network* (BN) for $P$ if $P$ factorizes according to $G$, i.e.

$$P(X) = \prod_{k=1}^{d} P(X_k \mid \mathrm{pa}(k)). \tag{1}$$

Obviously, by the chain rule of probability, a BN is not necessarily unique—any permutation of the variables can be used to construct a valid BN according to (1). A *minimal I-map* of $P$ is any BN such

that removing any edge would violate (1). The importance of the factorization (1) is that it implies that separation in $G$ implies conditional independence in $P$, and a minimal I-map encodes as many such independences as possible (although not necessarily all independences). For more details, a review of relevant graphical modeling concepts is included, see Appendix B.

The purpose of structure learning is twofold: 1) To identify a unique BN $G$ which can be identified from $P$, and 2) To devise algorithms to learn $G$ from data. This is our main goal.

# 3 Identifiability of minimal I-maps

In this section we introduce our main assumption regarding identifiability of minimal I-maps of $P$.

## 3.1 Conditions for identifiability

For $X_k \in V \setminus A_{j+1}$, define $\mathrm{an}_j(k) := \mathrm{an}(X_k) \cap L_{j+1}$, i.e $\mathrm{an}_j(k)$ denotes the ancestors of $X_k$ in $L_{j+1}$. By convention, we let $A_0 = \emptyset$. Finally, with some abuse of notation, let $L(X_k) \in [r]$ indicate which layer $X_k$ is in, i.e. $L(X_k) = j$ if and only if $X_k \in L_j$.

**Condition 1.** For each $X_k \in V$ with $L(X_k) \geq 2$ and for each $j = 0, \cdots, L(X_k) - 2$, there exists $X_i \in \mathrm{an}_j(k)$ such that following two conditions hold:

(C1) $H(X_i \,|\, A_j) < H(X_k \,|\, A_j)$,

(C2) $I(X_k; X_i \,|\, A_j) > 0$.

We refer $X_i \in \mathrm{an}_j(k)$ satisfying (C1) and (C2) in Condition 1 as the *important ancestors* for $X_k$, denoted by $\mathrm{im}_j(k)$. Thus, another way of stating Condition 1 is that for each $X_k \in V$ with $L(X_k) \geq 2$ and for each $j = 0, \cdots, L(X_k) - 2$, there exists an important ancestor, i.e. $\mathrm{im}_j(k) \neq \emptyset$.

The idea behind this condition is the following: Suppose we wish to identify the $t$th layer of $G$. Condition 1 requires that for every node *after* $L_t$ (represented by $X_k$), there is at least one node $X_i$ in $L_t$ satisfying (C1) and (C2). The intuition is the entropy (uncertainty) in the entire system is required to increase as time evolves.

Condition (C1) is the operative condition: By contrast, (C2) is just a nondegeneracy condition that says $X_i \not\perp\!\!\!\perp X_k \,|\, A_j$, which will be violated only when $\mathrm{an}_j(k)$ is conditionally independent of $X_k$ given $A_j$. At the population level, (C2) is superficially similar to faithfulness, however, a closer look reveals significant differences. One example of conditional independence between an entire layer of ancestors and descendants is path-cancellation, where effects through *multiple* paths are neutralized through delicate choices of parameters, whereas unfaithfulness occurs when there happens to be just one such path cancellation. Moreover, (C2) only applies to a small set of ancestral sets, whereas faithfulness applies to all possible $d$-separating sets. Not only is this kind of path-cancellation for all of $\mathrm{an}_j(k)$ unlikely, we show in Appendix G that this is essentially the only way (C2) can be violated: If $G$ is a poly-forest, then (C2) always holds, see Lemma G.1.

In Section 3.2, we will discuss this condition in the context of an algorithm and an example, which should help explain its purpose better. Before we interpret this condition further, however, let us point out why this condition is important: It identifies $G$.

**Theorem 3.1.** *If there is a minimal I-map $G$ satisfying Condition 1, then $G$ is identifiable from $P$.*

In order to better understand Condition 1, let us first compare it to existing assumptions such as equal variances [19, 36]. Indeed, there is a natural generalization of the equal variance assumption to Shannon entropy:

(C3) $H(X_k \,|\, \mathrm{pa}(k)) \equiv h^*$ is the same for each node $k = 1, \ldots, d$.

One reason to consider entropy is due to the fact that every distribution with $\mathbb{E}X^c < \infty$ for some $c > 0$ has well-defined entropy, whereas not all distributions have finite variance. Though quoted here, we will not require (C3) in the sequel; it is included here merely for comparison. Indeed, the next result shows that this "equal entropy" condition is a special case of Condition 1:

**Lemma 3.2.** *Assuming (C2), (C3) implies (C1). Thus, the "equal entropy" condition implies Condition 1.*

---
**Algorithm 1** Learning DAG structure
---
**Input:** $X = (X_1, \ldots, X_d), \omega$
**Output:** $\widehat{G}$.

1. Initialize empty graph $\widehat{G} = \emptyset$ and $j = 0$
2. Set $\widehat{L}_j = \emptyset$, let $\widehat{A}_j = \cup_{t=0}^j \widehat{L}_t$
3. While $V \setminus \widehat{A}_j \neq \emptyset$:
    (a) For $k \notin \widehat{A}_j$, estimate conditional entropy $H(X_k \mid \widehat{A}_j)$ by some estimator $\widehat{h}_{jk}$.
    (b) Initialize $\widehat{L}_{j+1} = \emptyset$, $\widehat{S}_{j+1} = \emptyset$. Sort $\widehat{h}_{jk}$ in ascending order and let $\widehat{\tau}^{(0)}$ be the corresponding permutation of $V \setminus \widehat{A}_j$.
    (c) For $\ell \in 0, 1, 2, \ldots$ until $|\widehat{\tau}^{(\ell)}| = 0$: **[TAM step]**
        i. $\widehat{L}_{j+1} = \widehat{L}_{j+1} \cup \{\widehat{\tau}_1^{(\ell)}\}$.
        ii. For $k \notin \widehat{A}_j \cup \widehat{L}_{j+1} \cup \widehat{S}_{j+1}$, estimate $I(X_k; \widehat{\tau}_1^{(\ell)} \mid \widehat{A}_j)$.
        iii. Set $\widehat{S}_{j+1} = \widehat{S}_{j+1} \cup \{k : \widehat{I}_{jk}^{(\ell)} \geq \omega\}$.
        iv. $\widehat{\tau}^{(\ell+1)} = \widehat{\tau}^{(\ell)} \setminus \left( \widehat{L}_{j+1} \cup \widehat{S}_{j+1} \right)$
    (d) For $k \in \widehat{L}_{j+1}$, set $\mathrm{pa}_{\widehat{G}}(k) = \mathrm{MB}(X_k; \widehat{A}_j)$.
    (e) Update $j = j + 1$.
4. Return $\widehat{G}$.
---

In fact, (C1) significantly relaxes (C3). The latter implies that all nodes in $L_{j+1}$ have smaller conditional entropy than $X_k$, whereas (C1) only requires this inequality to hold for at least one ancestor $X_i \in \mathrm{an}_j(k)$. Moreover, something even stronger is true: The equal variance condition can be relaxed to *unequal* variances (see Assumption 1 in [20]), and we can derive a corresponding "unequal entropy" condition. This condition is also a special case of Condition 1. We can also construct explicit examples that satisfy Condition 1, but neither the equal nor unequal entropy condition. For details, see Appendix C.3.

*Remark* 1. Condition 1 can be relaxed even further: See Appendix C.2 and Remark 3 for a discussion along with its corresponding algorithm.

### 3.2 Algorithmic interpretation

The proof of Theorem 3.1 motivates a natural algorithm to learn the DAG, shown in Algorithm 1. This algorithm exploits the fact that given $A_j$, nodes within $L_{j+1}$ are mutually independent. Algorithm 1 is a layer-wise DAG learning algorithm. For each layer, it firstly sorts the conditional entropies in ascending order $\tau$, then runs a "Testing and Masking" (TAM) step to distinguish nodes in $L_{j+1}$ from remaining ones ($X_k$): We use $\mathrm{im}_j(k)$ defined in (C1) to detect and *mask* $X_k \notin L_{j+1}$ by *testing* conditional independence. By masking, we mean we do not consider the nodes being masked when proceeding over the entropy ordering $\tau$ to identify $L_{j+1}$.

In order to see how Algorithm 1 works, consider the example shown in Figure 1(a). In the first step with $j = 0$, we use marginal entropy (i.e. since $A_0 = \emptyset$) to distinguish $L_1$ from the remaining nodes. Let $H(X_\ell) := h_\ell$ and assume for simplicity that the nodes are ordered such that $h_1 < h_2 < \cdots < h_6$ (Step 3(b)). Apparently, the inequalities that $h_3 < h_{[4:6]}$ and $h_5 < h_6$ imply (C3) does not hold here. Suppose there are no spurious edges, i.e. descendants and ancestors are dependent. Now we can see conditions in Theorem 3.1 are satisfied and the important ancestors for $X_2, X_3, X_5$ are $X_1, X_1, X_4$ respectively. The implementation of Algorithm 1 is visualized in Figure 1(b): In the first loop, $X_1$ is taken into $L_1$ and $X_2, X_3$ are masked due to dependence (Step 3c(iii)). In the second loop, $X_4$ is added to $L_1$ and then $X_5$ is masked. Finally, with $X_6$ put into $L_1$, we have identified $L_1$.

It is worth emphasizing that the increasing order of marginal entropies in this example is purely for simplicity of presentation, and does not imply any information on the causal order of the true DAG. The marginal entropies of nodes need not be monotonic with respect to the topological order of $G$.

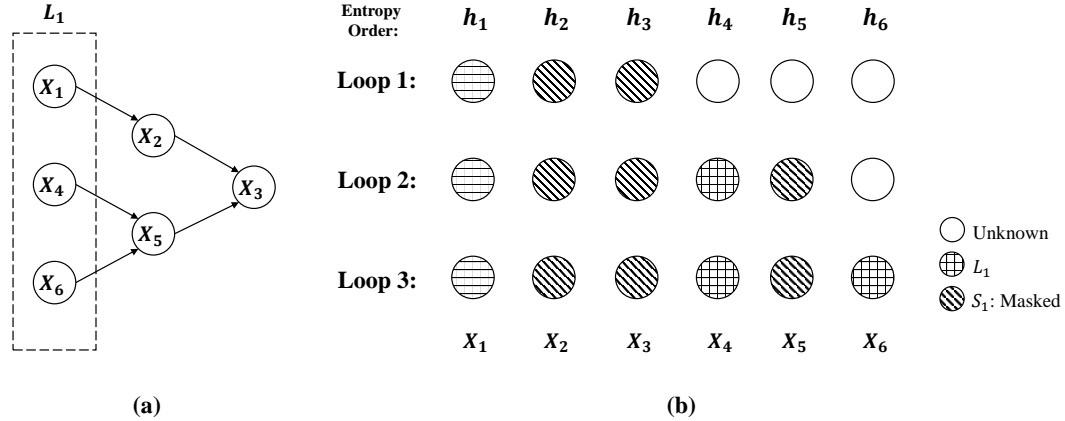

Figure 1: Example for TAM algorithm: (a) True DAG with $L_1$ outlined; (b) Status of the algorithm after each loop, denoted by different patterns of nodes.

---

**Algorithm 2** Possible Parental Set (PPS) procedure

---

**Input:** $X = (X_1, \ldots, X_d)$, $k$, $A$, $\kappa$

**Output:** Conditional entropy $\widehat{h}$, Markov boundary $\widehat{m}$

1. Initialize $\widehat{m} = \emptyset$, loop until $\widehat{m}$ does not change:
   (a) For $\ell \in A \setminus \widehat{m}$, estimate conditional mutual information $I(X_\ell; X_k \mid \widehat{m})$ by some estimator $\widehat{I}_\ell$.
   (b) Let $\ell^* = \arg\max_{\ell \notin A \setminus \widehat{m}} \widehat{I}_\ell$, if $\widehat{I}_{\ell^*} > \kappa$, set $\widehat{m} = \widehat{m} \cup \{\ell^*\}$.
2. Estimate conditional entropy $H(X_k \mid \widehat{m})$ by some estimator $\widehat{h}$.
3. Return conditional entropy estimation $\widehat{h}$ and estimated Markov boundary $\widehat{m}$.

---

## 4 Local Markov boundary search

Algorithm 1 assumes that we can learn $\mathrm{MB}(X_k; A_j)$, the Markov boundary of $X_k$ in the ancestral set $A_j$. This is a well-studied problem in the literature, and a variety of greedy algorithms have been proposed for learning Markov boundaries from data [1, 16, 18, 34, 41, 42], all based on the same basic idea: Greedily add variables whose association with $X_k$ is the highest. In this section, we establish theoretical guarantees for such a greedy algorithm. In the next section, we apply this algorithm to reconstruct to full DAG $G$ via Algorithm 1.

To present our Markov boundary search algorithm, we first need to set the stage. Let $X_k \in V$ be any node and $A$ an ancestral set of $X_k$. We wish to compute $\mathrm{MB}(X_k; A)$ and $H(X_k \mid A)$. An algorithm for this is outlined in Algorithm 2. In contrast to many existing local search methods for learning Markov boundaries, this algorithm is guaranteed to return the parents of $X_k$ in $G$. More specifically, if $X_k \in L_{j+1}$, then $\mathrm{MB}(X_k; A_j) = \mathrm{pa}(X_k)$. For this reason, we refer to Algorithm 2 as *possible parent selection*, or PPS for short. In fact, PPS is exactly the forward phase of the well-known IAMB algorithm for Markov blanket discovery [42] with conditional mutual information used both as an association measure and as a conditional independence test.

Although PPS will always return a valid Markov *blanket*, without a backward phase to remove unnecessary variables added by the forward greedy step, PPS may fail to return a *minimal* Markov blanket, i.e. the Markov boundary. The following condition is enough to ensure no unnecessary variables are included:

**Condition 2** (PPS condition). For any proper subset $m \subsetneq \mathrm{MB}(X_k; A)$ and any node $X_\ell \in A \setminus \mathrm{MB}(X_k; A)$, there exists $X_c \in \mathrm{MB}(X_k; A) \setminus m$ such that

$$I(X_k; X_c \mid m) > I(X_k; X_\ell \mid m).$$

Condition 2 requires that nodes in $\mathrm{MB}(X_k; A)$ always contribute larger conditional mutual information marginally than those that are not in $\mathrm{MB}(X_k; A)$. Thus when we do greedy search to select parents in Algorithm 2, only the nodes in $\mathrm{MB}(X_k; A)$ will be chosen. Therefore, with a proper threshold $\kappa$, this set can be perfectly recovered without incorporating any nuisance nodes.

We now present the sample complexity of using PPS to recover Markov boundaries under Condition 2. We will make frequent use of the Markov boundary and its size:

$$m_{Ak} := \mathrm{MB}(X_k; A), \quad M_{Ak} := |m_{Ak}|. \tag{2}$$

In particular, by definition we have $X_k \perp\!\!\!\perp (A \setminus m_{Ak}) \,|\, m_{Ak}$. Under Condition 2, we further define following quantities:

$$\widetilde{\Delta}_{Ak} := \min_{m \subsetneq m_{Ak}} \left[ \max_{c \in m_{Ak} \setminus m} I(X_k; X_c \,|\, m) - \max_{\ell \in A_j \setminus m_{Ak}} I(X_k; X_\ell \,|\, m) \right] > 0,$$

$$\xi_{Ak} := \min_{m \subsetneq m_{Ak}} \min_{c \in m_{Ak} \setminus m} I(X_k; X_c \,|\, m)/2 > 0. \tag{3}$$

$\widetilde{\Delta}_{Ak}$ is the gap between the mutual information of nodes inside or outside of $m_{Ak}$. $\xi_{Ak}$ is the minimum mutual information contributed by the nodes in $m_{Ak}$. The larger these quantities, the easier the $\mathrm{MB}(X_k; A)$ is to be recovered.

**Proposition 4.1.** *Fix $k \in V$ and let $A$ be any set of ancestors of $X_k$. Suppose Condition 2 holds. Algorithm 2 is applied for to estimate $m_{Ak}$ and $H(X_k \,|\, A)$ with $\kappa \leq \xi_{Ak}$, we have with $t \leq \min\left(\kappa, \tilde{\Delta}_{Ak}/2\right)$*

$$P\left(\widehat{m}_{Ak} = m_{Ak}, \left|\widehat{H}(X_k \,|\, A) - H(X_k \,|\, A)\right| < t\right) \geq 1 - (M_{Ak} + 2)|A| \frac{\delta_{M_{Ak}}^2}{t^2}.$$

*where $\delta_{M_{Ak}}^2$ is the estimation error of conditional entropy defined by* (9) *in Appendix E.1, which depends on $M_{Ak}$.*

A naïve analysis of this algorithm would have a sample complexity that depends on the size of the ancestral set $A$; note that our more fine-grained analysis depends instead on the size of the Markov boundary $\mathrm{MB}(X_k; A)$. We assume with sample size large enough, $\delta_{M_{Ak}}^2$ is small such that the right hand side remains to be positive and goes to $1$. The proof of Proposition 4.1 is deferred to Appendix E.2.

Condition 2 ensures the success of the greedy PPS algorithm. Although this assumption is not strictly necessary for structure learning (see Appendix C.5 for details), it significantly improves the sample complexity of the structure learning algorithm (Algorithm 1). Thus, it is worthwhile to ask when Condition 2 holds: We will take up this question again in Section 5.3.

## 5 Learning DAGs

Thus far, we have accomplished two important subtasks for learning a DAG: In Section 3, we identified its layer decomposition $L = (L_1, \ldots, L_r)$. In Section 4, we showed that the PPS procedure successfully recovers (ancestral) Markov boundaries. Combining these steps, we obtain a complete algorithm for learning $G$. In this section, we study the computational and sample complexity of this algorithm; proofs are deferred to the appendices.

We adopt the notations for Markov boundaries as in (2):

$$m_{jk} := \mathrm{MB}(X_k; A_j) \quad M_{jk} := |m_{jk}| \tag{4}$$

Therefore, $X_k \perp\!\!\!\perp (A_j \setminus m_{jk}) \,|\, m_{jk}$. A critical quantity in the sequel will be the size of the largest Markov boundary $m_{jk}$, which we denote by $M$:

$$M := \max_{jk} M_{jk} = \max_{jk} |m_{jk}|. \tag{5}$$

This quantity depends on the number of nodes $d$ and the structure of the DAG. For example, if the maximum in-degree of $G$ is 1, then $M = 1$. A related quantity that appears in existing work is the size of the largest Markov boundary *relative to all of $X$*, which may be substantially larger than $M$. The former quantity includes both ancestors *and* descendants, whereas $m_{jk}$ only contains ancestors. Analogously, let $M_j = \max_k M_{jk}$.

## 5.1 Algorithm

By combining Algorithms 1 and 2, we obtain a complete algorithm for learning the DAG $G$, which we refer to as the TAM algorithm, short for *Testing and Masking*. It consists of two parts:

1. Learning the layer decomposition $(L_1, \ldots, L_r)$ by minimizing conditional entropy and TAM step (Algorithm 1);
2. Learning the parental sets and reducing the size of conditioning sets by learning the Markov boundary $m_{jk} = \mathrm{MB}(X_k; A_j)$ (Algorithm 2).

Specifically, we use PPS (Algorithm 2) to estimate the conditional entropies (Step 3(a)), conditional mutual information (Step 3c(ii)), and the Markov boundary (Step 3(d)). For completeness, the complete procedure is detailed in Algorithm 3 in the supplement. More generally, Algorithm 2 can be replaced with any Markov boundary recovery algorithm or conditional entropy estimator; this highlights the utility of Algorithm 1 as a separate meta-algorithm for DAG learning.

One missing piece is the choice of estimators for conditional entropy and mutual information in steps (1a) and (2) of the PPS procedure. We adopt the minimax entropy estimator from Wu and Yang [47] by treating (without loss of generality) the joint entropy as the entropy of a multivariate discrete variable, although other estimators can be used without changing the analysis. The complexity of this estimator is exponential in $M$ (i.e. since there are up to $2^M$ states to sum over in any Markov boundary $m_{jk}$), so the computational complexity of Algorithm 2 is $O(Md2^M)$. In addition, for the TAM step, there are at most $\max_j d_j$ nodes in each layer to estimate conditional mutual information with remaining at most $d$ nodes, thus this step has computational complexity $O(d\max_j d_j 2^M)$. Thus assuming $M \lesssim \log d$, the overall computational complexity of Algorithm 1 is at most $O(r \times (d \times Md2^M + d \times \max_j d_j 2^M)) = O(d^3 r \log d)$. Specifically, for $r$ layers, we must estimate the conditional entropy of at most $d$ nodes, and call TAM step once.

## 5.2 Main statistical guarantees

In order to analyze the sample complexity of Algorithm 3 under Conditions 1 and 2, we introduce following positive quantities:

$$\Delta := \min_j \min_{k \in V \setminus A_{j+1}} \left( H(X_k \mid A_j) - H(X_i \mid A_j) \right) > 0$$

$$\eta := \min_j \min_{k \in V \setminus A_{j+1}} I(X_k; X_i \mid A_j) > 0$$

where $X_i \in \mathrm{im}_j(k)$ is defined in Condition 1. These two quantities are corresponding to the two conditions (C1) and (C2), which are used to distinguish each layer with its descendants. Compared to **strong** faithfulness, which is needed on finite samples, we only require a much smaller, restricted set of information measures to be bounded from zero. We also adopt the quantities defined in (3) by setting $A = A_j$ and drop the notation $A$ such that $\widetilde{\Delta}_{jk} := \widetilde{\Delta}_{A_j k}$ and $\xi_{jk} := \xi_{A_j k}$.

Finally we are ready to state the main theorem about sample complexity of Algorithm 1:

**Theorem 5.1.** *Suppose $P$ satisfies Conditions 1 and 2, and let $G$ be the minimal I-map identified by Theorem 3.1. Let $\widehat{G}$ be output of Algorithm 3 applied with $\omega \leq \eta/2$ and $\kappa \leq \min_{jk} \xi_{jk}$. Denote $\Delta_{\omega,\kappa}^* = \min_{jk}(\Delta/2, \omega, \kappa, \widetilde{\Delta}_{jk}/2)$. If $M \lesssim \log d$ and*

$$n \gtrsim \frac{d^2 r \log^3 d}{(\Delta_{\omega,\kappa}^*)^2 \epsilon},$$

*then $\widehat{G} = G$ with probability $1 - \epsilon$.*

Up to log factors, the sample complexity scales quadratically with the number of nodes $d^2$ and linearly in the depth $r \leq d$. In the worst case, this is cubic in the dimension. For example, if $G$ is a Markov chain then $M = 1$ and $r = d$, thus it suffices to have $n = \Omega(d^3 \log^3 d)$. Comparatively, most of previous work [6, 21, 44] only consider linear or parametric models. One recent work that provides nonparametric guarantees without assuming faithfulness is [17], who show that in general, $\Omega((dr/\epsilon)^{1+d/2})$ samples suffice to recover the topological ordering under an equal variance

assumption similar in spirit to (C3). Unlike the current work, which considers exact recovery of the full graph $G$, [17] does not include the reduction to Markov boundary search that is crucial to our exact recovery results.

In fact, as the proof indicates, the sample complexity of our result also depends exponentially on $M$. This explains the assumption that $M \lesssim \log d$; the logarithmic assumption is analogous to sparsity and results from not making any parametric assumptions on the model. Since our setting is fully nonparametric, exponential rates in the dimension $d$ are to be expected. Under stronger parametric assumptions, these exponential rates can likely be avoided. A detailed analysis of the dependency on $M$ can be found in the proofs, which are deferred to Appendix E.

Finally, in practice the quantities $\eta, \xi_{jk}$ needed in Theorem 5.1 may not be known, which hinders the choice of tuning parameters $\omega, \kappa$. In Appendix E.4 (see Theorem E.2) we discuss the selection of these tuning parameters in a data-dependent way.

### 5.3 Application to learning polytrees

Learning polytrees is one of the simplest DAG learning problems. This problem was introduced in Rebane and Pearl [37] and in a seminal work, Dasgupta [10] showed that learning polytrees is NP-hard in general. In this section, we note that when the underlying DAG is a polytree (or more generally, a polyforest), Condition 2 is always satisfied, and therefore we have an efficient algorithm for learning identifiable polyforests that satisfy Condition 1.

Recall that a polyforest is a DAG whose skeleton (i.e. underlying undirected graph) has no cycles.

**Theorem 5.2.** *If $P$ satisfies* (1) *for some polyforest $G$, then Condition 2 is satisfied.*

The detailed proof can be found in Appendix F. As a result, it follows immediately (by combining Theorems 5.1 and 5.2 with Lemma G.1) that any polytree satisfying (C1) is learnable.

The crucial property of a poly-forest used in proving Theorem 5.2 is that there exists at most one directed path between any two nodes in the graph. However, the existence of multiple directed paths between two nodes does not necessarily imply that Condition 2 will fail: There are concrete examples of graphs satisfying Condition 2 with arbitrarily many paths between two nodes. An example is given by the DAG $G = (V, E)$ with $V = (Z, X_1, \ldots, X_k, Y)$ such that $Z \to X_i \to Y$ for each $i$. Thus, this assumption holds more broadly than suggested by Theorem 5.2. It is an interesting problem for future work to study this further.

## 6 Experiments

We conduct a brief simulation study to demonstrate the performance of Algorithm 1 and compare against some common baselines: PC [39], GES [7]. We focus on the fully discrete setting. All implementation details can be found in Appendix I. The code implementing TAM algorithm is available at `https://github.com/MingGao97/TAM`. We stress that the purpose of these experiments is simply to illustrate that the proposed algorithm can be implemented in practice, and successfully recovers the edges in $G$ as predicted by our theoretical results.

We evaluate the performance of aforementioned algorithms by Structural Hamming distance (SHD): This is a standard metric for DAG learning that counts the total number of edge additions, deletions, and reversals needed to convert the estimated graph into the reference one. Since PC and GES both return a CPDAG that may contain undirected edges, we evaluate these algorithms favourably by assuming correct orientation for undirected edges wherever they are present.

We simulate DAGs from three graph types: Poly-trees (Tree) , Erdös-Rényi (ER), and Scale-Free (SF) graphs. As discussed in Section 5.3, poly-tree models are guaranteed to satisfy Condition 2, whereas general DAGs such as ER or SF graphs are not, so this provides a test case for when this condition may fail. We generate data according to two models satisfying the "equal entropy" condition (C3). As discussed in Appendix C.3, (C3) implies our main identifiability Condition 1.

- "Mod" model (MOD): $X_k = (S_k \mod 2)^{Z_k} \times (1 - (S_k \mod 2))^{1-Z_k}$ where $S_k = \sum_{\ell \in \mathrm{pa}(k)} X_\ell$ with $Z_k \sim \mathrm{Ber}(0.2)$
- Additive model (ADD): $X_k = \sum_{\ell \in \mathrm{pa}(k)} X_\ell + Z_k$ with $Z_k \sim \mathrm{Ber}(0.2)$

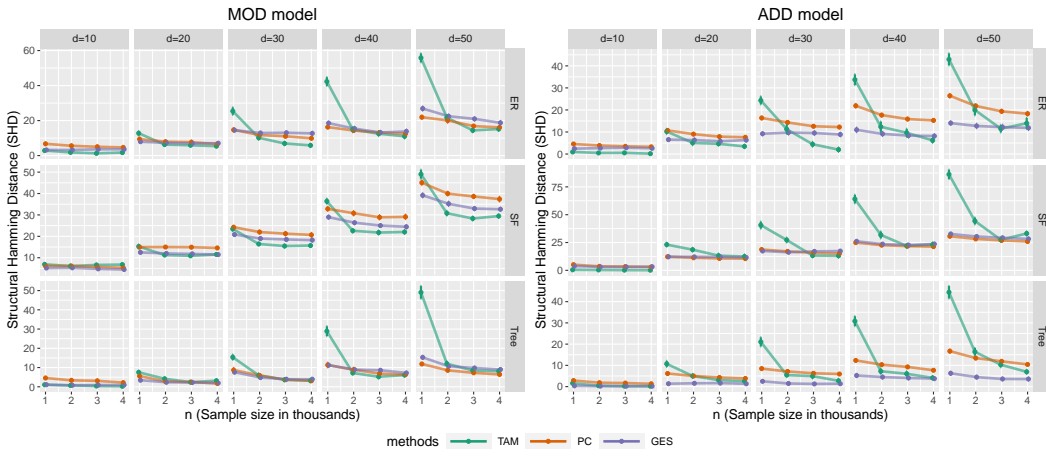

Figure 2: SHD vs sample size $n$ for different dimensions and graph types. Left panel is for MOD model; Right panel is for ADD model.

Figure 2 (left, right) illustrates the performance in terms of structural Hamming distance (SHD) between the true graph and the estimated graph. As expected, our algorithm successfully recovers the underlying DAG $G$ and performs comparably to PC and GES, which also perform quite well. We stress that the current experiments are used for simple illustration thus not well-optimized compared to existing algorithms or fully exploited the main condition either. The relatively good performance of PC/GES partially comes from the fact that our synthetic models are all in fact faithful, see Appendix C.7 for further discussion.

# 7  Conclusion

The main goal of this paper has been to better understand the underlying assumptions required for DAG models to be estimated from data. To this end, we have provided a new identifiability result along with a learning algorithm, which turns out to generalize existing ones, and analyzed a greedy, local search algorithm for discovering Markov boundaries. This local search algorithm can be used to provably learn the structure of a minimal I-map in polynomial time and sample complexity as long as the Markov boundaries are not too large. Nonetheless, there are many interesting directions for future work. Perhaps the most obvious is relaxing the logarithmic dependence on $d$ in $M$. It would also interesting to investigate lower bounds on the sample complexity of this model, as well as additional identifiability conditions.

**Acknowledgements**

We thank the anonymous reviewers for their helpful comments in improving the manuscript. B.A. was supported by NSF IIS-1956330, NIH R01GM140467, and the Robert H. Topel Faculty Research Fund at the University of Chicago Booth School of Business. All statements made are solely due to the authors and have not been endorsed by the NSF.

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
