# Efficient Bayesian network structure learning via local Markov boundary search—Supplement

## A    Complete algorithm description

For completeness and reproducibility, the full TAM algorithm combining Algorithms 1 and 2 is detailed in Algorithm 3.

---

**Algorithm 3** TAM algorithm for learning DAGs

---

**Input:** $X = (X_1, \ldots, X_d), \omega, \kappa$
**Output:** $\widehat{G}$.

1. Initialize empty graph $\widehat{G} = \emptyset$ and $j = 0$.

2. Set layer $\widehat{L}_0 = \emptyset$, let $\widehat{A}_j = \cup_{t=0}^{j} \widehat{L}_t$.

3. While $V \setminus \widehat{A}_j \neq \emptyset$:

    (a) For $k \notin \widehat{A}_j$, apply PPS$(X, k, \widehat{A}_j, \kappa)$ (See Algorithm 2) to obtain the estimated Markov boundary $\widehat{m}_{jk}$ along with an estimate $\widehat{h}_{jk}$ of the corresponding conditional entropy $H(X_k \mid \widehat{A}_j)$.

    (b) Initialize $\widehat{L}_{j+1} = \emptyset$, $\widehat{S}_{j+1} = \emptyset$. Sort $\widehat{h}_{jk}$ in ascending order and let $\widehat{\tau}^{(0)}$ be the corresponding permutation of $V \setminus A_j$.

    (c) For $\ell \in 0, 1, 2, \ldots$ until $|\widehat{\tau}^{(\ell)}| = 0$: **[TAM step]**

        i. $\widehat{L}_{j+1} = \widehat{L}_{j+1} \cup \{\widehat{\tau}_1^{(\ell)}\}$.
        ii. For $k \notin \widehat{A}_j \cup \widehat{L}_{j+1} \cup \widehat{S}_{j+1}$, estimate $I(X_k; \widehat{\tau}_1^{(\ell)} \mid \widehat{m}_{jk})$ by some estimator $\widehat{I}_{jk}^{(\ell)}$
        iii. Set $\widehat{S}_{j+1} = \widehat{S}_{j+1} \cup \{k : \widehat{I}_{jk}^{(\ell)} \geq \omega\}$.
        iv. $\widehat{\tau}^{(\ell+1)} = \widehat{\tau}^{(\ell)} \setminus \left( \widehat{L}_{j+1} \cup \widehat{S}_{j+1} \right)$

    (d) For $k \in \widehat{L}_{j+1}$, set $\mathrm{pa}_{\widehat{G}}(k) = \widehat{m}_{jk}$.

    (e) Update $j = j + 1$.

4. Return $\widehat{G}$.

---

## B    Graphical model background

In this appendix, we recall some basic facts regarding graphical models that are used throughout the proofs. This section will also help to explain the importance of the positivity assumption on $P$, as well as the concept of faithfulness. For more background on graphical models, see Koller and Friedman [29], Lauritzen [31].

**Uniqueness of Markov boundaries**    The *Markov blanket* of a node $X_k$ relative to some subset $S \subset V$ is any subset $m \subset S$ such that $X_k \perp\!\!\!\perp (S \setminus m) \mid m$. A Markov boundary is a minimal Markov blanket, i.e. a Markov blanket $m$ such that no proper subset of $m' \subsetneq m$ satisfies $X_k \perp\!\!\!\perp (S \setminus m') \mid m'$. Neither the Markov blanket nor the Markov boundary are unique in general. A key fact regarding Markov boundaries is that when $P$ is strictly positive, they are unique. Recall that we denote the Markov boundary of $X_k$ relative to $S$ by $\mathrm{MB}(X_k; S)$.

**Lemma B.1.** *If $P(X = x) > 0$ for all $x \in \{0, 1\}^d$, then for any $S \subset V$, the Markov boundary* $\mathrm{MB}(X_k; S)$ *exists and is unique.*

For a direct proof, see Proposition 3.1.3 in Drton et al. [13]. This lemma remains true if $P$ is replaced by a (strictly positive) density function. More generally, Markov boundaries are unique as long as the *intersection property* of conditional independence holds in $P$ [11, 35].

**Minimal I-maps and orderings**  A minimal I-map of $P$ is any DAG $G = (V, E)$ such that the following conditions hold:

1. $P$ factorizes over $G$, i.e. (1) holds, and
2. If any edge is removed from $E$, then (1) is violated.

In general, minimal I-maps are not unique. Given an ordering $\prec$ of the variables, a minimal I-map can be constructed as follows [see e.g., 29, §3.4.1]: For each $k$, define $\mathrm{pa}(k)$ to be $\mathrm{MB}(X_k; \prec_k)$, where $\prec_k := \{j : X_j \prec X_k\}$. This procedure is well-defined as long as $\mathrm{MB}(X_k; \prec_k)$ is unique, which is guaranteed by Lemma B.1. An important consequence of this procedure is that once the layer decomposition of a minimal I-map $G$ is known, the full DAG $G$ can be recovered by performing local search. To see this, recall that the layers $L_j$ define canonical ancestral sets $A_j$ and replace $\mathrm{MB}(X_k; \prec_k)$ above with $\mathrm{MB}(X_k; A_j)$, where $j$ is the largest index such that $X_k \notin A_j$.

**Faithfulness and $d$-separation**  Throughout the proofs, we make use of the concept of $d$-separation defined below. Although faithfulness is never assumed, it is useful to recall its definition for completeness.

A *trail* in a directed graph $G$ is any sequence of distinct nodes $X_{i_1}, \ldots, X_{i_\ell}$ such that there is an edge between $X_{i_m}$ and $X_{i_{m+1}}$. The orientation of the edges does not matter. For example, $X \leftarrow Y \rightarrow Z$ would be a valid trail. A trail of the form $X_{i_{m-1}} \rightarrow X_{i_m} \leftarrow X_{i_{m+1}}$ is called a *v-structure*. We say a trail $t$ is *active* given another set $C$ if (a) $\mathrm{de}(X_{i_m}) \cap C \neq \emptyset$ for every $v$-structure $X_{i_{m-1}} \rightarrow X_{i_m} \leftarrow X_{i_{m+1}}$ in $t$ and (b) no other node in $t$ is in $C$. In other words, $t \cap C$ consists only of central nodes in some $v$-structure contained entirely in $t$.

**Definition 1.**  Let $A, B, C$ be three sets of nodes in $G$. We say that $A$ and $B$ are $d$-separated by $C$ if there is no active trail between any node $a \in A$ and $b \in B$ given $C$.

An important consequence of (1) is the following: If $G$ satisfies (1) for some $P$ and $A$ and $B$ are $d$-separated by $C$ in $G$, then $A \perp\!\!\!\perp B \mid C$ in $P$ (29, Theorems 3.1, 3.2, or §3.2.2 in 31).

Thus, if $G$ is a BN of $P$, then $d$-separation can be used to read off a subset of the conditional independence relations in $P$. Whenever the reverse implication holds—i.e. conditional independence in $P$ implies $d$-separation in $G$, we say that $P$ is *faithful* to $G$. This condition does not hold in general, not even for minimal I-maps.

*Remark* 2.  Faithfulness is a standard assumption in the literature on BNs. Assuming $G$ is faithful to $P$ ensures that the *Markov equivalence class* of $P$ is identified, however, this is not the same as identifying $G$. More precisely, faithfulness identifies a CPDAG, which is a partially directed graph that encodes the set of conditional independence relationships shared by every DAG in the Markov equivalence class. This can be a strong assumption, especially with finite samples [43]. By contrast, our approach is to circumvent faithfulness and impose assumptions that identify a bona-fide DAG $G$. See Appendix C.7 for an explicit example where Condition 1 identifies an unfaithful DAG.

## C  Extensions and further examples

The results in Sections 3-5 make several assumptions that are not strictly necessary. In this appendix, we briefly outline how these assumptions can be relaxed. In Appendix C.1 we show how the positivity assumption can be replaced by a slightly weaker nondegeneracy condition. In Appendix C.2-C.3 we discuss how Condition 1 can be relaxed and how it is compared with existing identifiability results. Then we discuss examples and extensions of Condition 2 in Appendices C.4-C.5. Finally, in Appendix C.6 we discuss extensions to more general distributions.

### C.1  Positivity and a nondegeneracy condition

Throughout the paper, we have assumed that $P$ is strictly positive. In fact, all of the results will go through under the following slightly weaker condition:

**Condition 3** (Nondegeneracy).  $I(X_k; \mathrm{pa}(k) \mid A) > 0$ for any ancestral set $A \subset [d]$ such that $\mathrm{pa}(k) \setminus A \neq \emptyset$.

This condition implies that there is still some information between $X_k$ and $\mathrm{pa}(k)$, even after learning everything about $A$. This is quite reasonable: If $\mathrm{pa}(k) \setminus A \neq \emptyset$ is nonempty, then there is still at least

one parent of $X_k$ unaccounted for after conditioning on $A$. This "missing parent" accounts for the "missing mutual information" that makes this quantity positive. Seemingly reasonable, there are some degenerate cases where Condition 3 may not hold. The following lemma makes the characterization of nondegeneracy more precise:

**Lemma C.1.** *Suppose that for any two disjoint subsets $A, B \subset X$, $P(A \,|\, B) \notin \{0,1\}$. Then Condition 3 holds for any minimal I-map of $P$.*

This lemma shows that as long as the dependencies implied by $G$ are non-deterministic, Condition 3 will always be satisfied. In fact, this is guaranteed by the positivity of $P$, which we have assumed already:

**Corollary C.2.** *If $P$ is strictly positive, then Condition 3 holds for any minimal I-map of $P$.*

Before proving Lemma C.1, we first show how Corollary C.2 follows as a consequence. Indeed, this follows immediately from Lemma C.1 and the following lemma:

**Lemma C.3.** *If $P(X = x) > 0$ for all $x \in \{0,1\}^d$, then for any two disjoint subsets $A, B \subset X$, $P(A \,|\, B) \notin \{0,1\}$.*

*Proof.* For any $a, b$, Bayes' rule implies $P(A = a \,|\, B = b) > 0$. Now we show $P(A = a \,|\, B = b) \neq 1$. Suppose the contrary, then

$$P(A = a, B = b) = P(B = b) = \sum_{a'} P(A = a', B = b) \implies \sum_{a' \neq a} P(A = a', B = b) = 0,$$

which is contradictory to $P(A = a, B = b) > 0$ and completes the proof. $\square$

Now we prove Lemma C.1.

*Proof.* (Proof of Lemma C.1) We proceed by contradiction. Suppose $I(X_k; \mathrm{pa}(k) \,|\, A)) = 0$, then

$$X_k \perp\!\!\!\perp \mathrm{pa}(k) \,|\, A$$

Let $\mathrm{pa}_1(k) = \mathrm{pa}(k) \cap A$, $\mathrm{pa}_2(k) = \mathrm{pa}(k) \backslash A$, and $A' = A \backslash \mathrm{pa}_1(k)$, so that $\mathrm{pa}(k) = \mathrm{pa}_1(k) \cup \mathrm{pa}_2(k)$, $A = \mathrm{pa}_1(k) \cup A'$, and $\mathrm{pa}_2(k) \cap A = \emptyset$, $\mathrm{pa}_2(k) \neq \emptyset$. Therefore,

$$P(X_k \,|\, \mathrm{pa}_1(k), A')P(\mathrm{pa}_1(k), \mathrm{pa}_2(k) \,|\, \mathrm{pa}_1(k), A') = P(X_k, \mathrm{pa}_1(k), \mathrm{pa}_2(k) \,|\, \mathrm{pa}_1(k), A').$$

Since

$$P(\mathrm{pa}_2(k) = y_2 \,|\, \mathrm{pa}_1(k) = y_1', A' = a') = \sum_{y_1} P(\mathrm{pa}_1(k) = y_1, \mathrm{pa}_2(k) = y_2 \,|\, \mathrm{pa}_1(k) = y_1', A' = a')$$

$$= \sum_{y_1 = y_1'} P(\mathrm{pa}_1(k) = y_1, \mathrm{pa}_2(k) = y_2 \,|\, \mathrm{pa}_1(k) = y_1', A' = a')$$

$$= P(\mathrm{pa}_1(k) = y_1', \mathrm{pa}_2(k) = y_2 \,|\, \mathrm{pa}_1(k) = y_1', A' = a').$$

Thus we can simplify to

$$P(X_k \,|\, \mathrm{pa}_1(k), A')P(\mathrm{pa}_2(k) \,|\, \mathrm{pa}_1(k), A') = P(X_k, \mathrm{pa}_2(k) \,|\, \mathrm{pa}_1(k), A')$$

which amounts to

$$X_k \perp\!\!\!\perp \mathrm{pa}_2(k) \,|\, (A', \mathrm{pa}_1(k)).$$

Combined with the Markov property $X_k \perp\!\!\!\perp A' \,|\, (\mathrm{pa}_1(k), \mathrm{pa}_2(k))$ (i.e. since $A' \subset \mathrm{nd}(k)$), we have

$$P(X_k, \mathrm{pa}_2(k), A' \,|\, \mathrm{pa}_1(k)) = P(X_k, \mathrm{pa}_2(k) \,|\, A', \mathrm{pa}_1(k))P(A' \,|\, \mathrm{pa}_1(k))$$

$$= P(X_k \,|\, A', \mathrm{pa}_1(k))P(\mathrm{pa}_2(k), A' \,|\, \mathrm{pa}_1(k))$$

$$= P(X_k, A' \,|\, \mathrm{pa}_1(k), \mathrm{pa}_2(k))P(\mathrm{pa}_2(k) \,|\, \mathrm{pa}_1(k))$$

$$= P(X_k \,|\, \mathrm{pa}_2(k), \mathrm{pa}_1(k))P(\mathrm{pa}_2(k), A' \,|\, \mathrm{pa}_1(k)).$$

By Condition 3, $P(\mathrm{pa}_2(k), A' \,|\, \mathrm{pa}_1(k)) \notin \{0,1\}$, so that

$$P(X_k \,|\, A', \mathrm{pa}_1(k)) = P(X_k \,|\, \mathrm{pa}_2(k), \mathrm{pa}_1(k))$$

holds for different combinations of $(A', \mathrm{pa}_2(k))$. These are two functions of $A'$ and $\mathrm{pa}_2(k)$ given $\mathrm{pa}_1(k)$, and are equal for all possible combinations of $(A', \mathrm{pa}_2(k))$, i.e. it is independent of what values they take on. It follows that

$$P(X_k \mid \mathrm{pa}_1(k)) = P(X_k \mid \mathrm{pa}_2(k), \mathrm{pa}_1(k)).$$

Since $G$ is an I-map, the joint probability factorizes over it

$$
\begin{aligned}
P(X_1, \cdots, X_d) &= \prod_{\ell=1}^{d} P(X_\ell \mid \mathrm{pa}(\ell)) \\
&= P(X_k \mid \mathrm{pa}_1(k), \mathrm{pa}_2(k)) \prod_{\ell \neq k} P(X_\ell \mid \mathrm{pa}(\ell)) \\
&= P(X_k \mid \mathrm{pa}_1(k)) \prod_{\ell \neq k} P(X_\ell \mid \mathrm{pa}(\ell)).
\end{aligned}
$$

Therefore we can remove the edges from $\mathrm{pa}_2(k)$ to $X_k$, which contradicts the minimality of $G$. The proof is complete. $\qquad\square$

## C.2 More general version of Condition 1

According to Lemma G.1, one sufficient condition for (C2) is to have no path-cancellation between $X_i$ and $X_k$. This can be further relaxed by following Theorem C.4 and Algorithm 4. For any ancestor $X_i \in \mathrm{an}_j(k)$, denote the subset $\mathrm{an}_j^i(k) \subseteq \mathrm{an}_j(k) \setminus X_i$ to be the ancestors with smaller conditional entropies than $X_i$, namely,

$$\mathrm{an}_j^i(k) := \{ X_\ell \in \mathrm{an}_j(k) \setminus X_i : H(X_\ell \mid A_j) \leq H(X_i \mid A_j) \}$$

**Theorem C.4.** *For each $X_k \in V$ with $L(X_k) \geq 2$, and for each $j = 0, \cdots, L(X_k) - 2$, there exists $X_i \in \mathrm{an}_j(k)$, which we refer as important ancestors $\mathrm{im}_j(k)$, such that following two conditions hold:*

*(U1).* $H(X_i \mid A_j) < H(X_k \mid A_j)$

*(U2).* $I(X_k; (\mathrm{an}_j^i(k), X_i) \mid A_j) > 0,$

*Then $G$ is identifiable from $P$.*

*Remark* 3. For any node $X_k$ not in current layer $L_{j+1}$, (U1) is exactly the same as (C1). (U2) requires its important ancestor together with other ancestors with smaller entropy in $L_{j+1}$ contribute to positive mutual information with $X_k$. Note that $\mathrm{an}_j^i(k)$ can be empty, in which case (U2) reduces to (C2). If this is applied to the equal entropy case in Condition (C3), (U2) is relaxed to

$$I(X_k; L_{j+1} \mid A_j) > 0$$

which requires the entropy of nodes not in $L_{j+1}$ to be conditional dependent with all nodes in $L_{j+1}$.

Algorithm 4 differs with Algorithm 1 in only one step. When testing independence in the TAM step, instead of estimating $I(X_k; \widehat{\tau}_1^{(\ell)} \mid \widehat{A}_j)$, we choose to estimate $I(X_k; \widehat{L}_{j+1} \mid \widehat{A}_j)$ to take advantage of $\mathrm{an}_j^i(k)$ to detect $X_k \notin L_{j+1}$. Since Theorem 3.1 and Algorithm 1 are special cases of Theorem C.4 and Algorithm 4, we only show the proof of the later (more general) theorem and the correctness of the later algorithm, which are shown in Appendix D.

## C.3 Comparison with "equal entropy" condition

The "equal entropy" Condition (C3) has a straightforward relaxation which follows directly along similar lines as Ghoshal and Honorio [19]. For completeness, we quote this result below; the proof is identical to this prior work and hence omitted. Denote $h_k = H(X_k \mid \mathrm{pa}(k))$.

**Condition 4.** There exists a topological ordering $\tau$ such that for all $j \in [d]$ and $\ell \in \tau_{[j+1:d]}$, the following holds:

1. If $k = \tau_j$ and $\ell$ are not in the same layer, then

$$h_k < h_\ell + I(X_\ell; \mathrm{pa}(\ell) \setminus \tau_{[1:j-1]} \mid X_{\tau_{[1:j-1]}}). \tag{6}$$

---
**Algorithm 4** TAM algorithm (general version)
---
**Input:** $X = (X_1, \ldots, X_d), \omega$
**Output:** $\widehat{L} = (\widehat{L}_1, \ldots, \widehat{L}_{\widehat{r}})$.

1. Initialize $\widehat{L}_0 = \emptyset$, let $\widehat{A}_j = \cup_{t=0}^{j} \widehat{L}_t$

2. For $j \in 0, 1, 2, \ldots$:

    (a) For $k \notin \widehat{A}_j$, estimate conditional entropy $H(X_k \mid \widehat{A}_j)$ by some estimator $\widehat{h}_{jk}$.

    (b) Initialize $\widehat{L}_{j+1} = \emptyset$, $\widehat{S}_{j+1} = \emptyset$. Sort $\widehat{h}_{jk}$ in ascending order and let $\widehat{\tau}^{(0)}$ be the corresponding permutation of $V \setminus A_j$.

    (c) For $\ell \in 0, 1, 2, \ldots$ until $|\widehat{\tau}^{(\ell)}| = 0$: **[TAM step]**

        i. Let $\widehat{L}_{j+1} = \widehat{L}_{j+1} \cup \{\widehat{\tau}_1^{(\ell)}\}$.

        ii. For $k \notin \widehat{A}_j \cup \widehat{L}_{j+1} \cup \widehat{S}_{j+1}$, estimate $I(X_k; \widehat{L}_{j+1} \mid \widehat{A}_j)$ by some estimator $\widehat{I}_{jk}^{(\ell)}$

        iii. Set $\widehat{S}_{j+1} = \widehat{S}_{j+1} \cup \{k : \widehat{I}_{jk}^{(\ell)} \geq \omega\}$.

        iv. $\widehat{\tau}^{(\ell+1)} = \widehat{\tau}^{(\ell)} \setminus \left( \widehat{L}_{j+1} \cup \widehat{S}_{j+1} \right)$

3. Return $\widehat{L} = (\widehat{L}_1, \ldots, \widehat{L}_{\widehat{r}})$.
---

2. If $k$ and $\ell$ are in the same layer, then either $h_k = h_\ell$ or (6) holds.

**Theorem C.5.** *If Condition 4 holds for some ordering $\tau$, then $\tau$ is identifiable from $P$.*

We can interpret Condition 4 in following two ways. As long as the conditional mutual information on the right side of (6) is positive, then the topological sort can be recovered by minimizing conditional entropies, much like the equal variance algorithm. On the other hand, compared to Condition (C3) which requires $h_k = h_\ell$, the conditional mutual information between child and parent nodes on the RHS is a bound on the difference $h_k - h_\ell$. Thus, Condition 4 can be interpreted as a relaxation of Condition (C3) in which the "violations" of equality are controlled by this conditional mutual information.

**Lemma C.6.** *Assuming (C2), Condition 4 implies Condition 1.*

*Proof.* Suppose $X_i = X_{\tau_s} \in \mathrm{an}_j(k)$ has the closest position to $X_k$ in terms of ordering $\tau$, then $X_{\tau_{[1:s-1]}}$ contains all the other ancestors of $X_k$ in $L_{j+1}$ except for $X_i$. Furthermore, $X_{\tau_{[1:s-1]}}$ also contains the ancestors of $\mathrm{an}_j(k)$ such that $P(X_k \mid A_j) = P(X_k \mid \text{ancestors of } \mathrm{an}_j(k))$. Therefore, we have (C1):

$$\begin{aligned} H(X_i \mid A_j) &= H(X_i \mid X_{\tau_{[1:s-1]}}) \\ &< H(X_k \mid X_{\tau_{[1:s-1]}}) \\ &= H(X_k \mid A_j, \mathrm{an}_j(k) \setminus X_i) \\ &\leq H(X_k \mid A_j) \end{aligned}$$

The first inequality is by (6), the second one uses the increasing property of conditional entropy when conditioning set is enlarged. □

*Remark* 4. Lemma C.6 implies Lemma 3.2. In fact, since Condition (C3) implies Condition 4, we have

$$\text{Condition (C3)} \implies \text{Condition 4} \implies \text{Condition 1}.$$

Finally, we conclude with an example to illustrate the difference between Condition 1 and Condition 4.

**Example 1.** Suppose the graph is

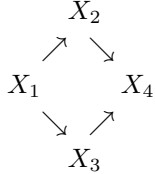

where

$$X_1 = Z_1, \quad Z_1 \sim Ber(0.2),$$
$$X_2 = -X_1 + Z_2 = -Z_1 + Z_2, \quad Z_2 \sim Ber(0.1),$$
$$X_3 = X_1 + Z_3 = Z_1 + Z_3, \quad Z_3 \sim Ber(0.2).$$

For $X_4$, we consider two models:

(M1) $X_4 = X_2 + Ber(\sigma(\epsilon X_3 + \beta_0))$

(M2) $X_4 = X_2 + X_3 + Z_4 = Z_2 + Z_3 + Z_4, \quad Z_4 \sim Ber(0.1),$

where $\sigma(x) = 1/(1 + \exp(-x))$. It is straightforward to check that each of these models does not satisfy equalities in Condition (C3), but satisfies either Condition 1 or Condition 4. Let's first look at the first three variables:

$$H(X_1) = h(0.2) \approx 0.500 \qquad H(X_2) \approx 0.733 \qquad H(X_3) \approx 0.778$$
$$H(X_2 \mid X_1) = h(0.1) \approx 0.325 < H(X_2) \quad H(X_3 \mid X_1) = h(0.2) \approx 0.500 < H(X_3)$$

So the subgraph on $(X_1, X_2, X_3)$ satisfies both Condition 1 and Conditions 4. Now consider the last node: For the first model (M1), $\beta_0 = \sigma^{-1}(0.1) = \log \frac{0.1}{1-0.1} \approx -0.219$. Choose $\epsilon$ small enough so that $\sigma(\epsilon X_3 + \beta_0) \approx 0.1$, e.g. if $\epsilon = 0.01$ then $\sigma(\epsilon X_3 + \beta_0) \in [0.1, 0.102]$ when $X_3 \in \{0, 1, 2\}$. By doing so,

$P(X_4 = -1) = 0.2 \times 0.9 \times 0.9$   $P(X_4 = -X_1 + 0 \mid X_1) = 0.9 \times 0.9$

$P(X_4 = 0) = 0.8 \times 0.9 \times 0.9 + 0.2 \times 0.9 \times 0.1 \times 2$   $P(X_4 = -X_1 + 1 \mid X_1) = 0.9 \times 0.1 \times 2$

$P(X_4 = 1) = 0.8 \times 0.9 \times 0.1 \times 2 + 0.2 \times 0.1 \times 0.1$   $P(X_4 = -X_1 + 2 \mid X_1) = 0.1 \times 0.1$

$P(X_4 = 2) = 0.8 \times 0.1 \times 0.1$   $H(X_4 \mid X_1) \approx 0.525 < H(X_4)$

$H(X_4) \approx 0.87$   $H(X_4 \mid X_1, X_2) \approx h(0.1) \approx 0.325$

  $< H(X_3 \mid X_1, X_2) < H(X_4 \mid X_1)$

Therefore (M1) satisfies (C1) and (C2) but not Condition 4, since the order between $X_3$ and $X_4$ is flipped. On the other hand, the second model (M2), where $X_1 \perp\!\!\!\perp X_4$, $I(X_1; X_4) = 0$, thus violates (C2) and will incorporate $X_4$ into the first layer. But Condition 4 still holds. Note that (M1) is also an example of path-cancellation, which is discussed in Lemma G.1.

## C.4 Examples of PPS condition

In this section, we illustrate some examples of DAGs satisfying Condition 2 for which there may exist multiple directed paths between two nodes. This shows that the poly-forest assumption in Theorem 5.2 is sufficient but not necessary for Condition 2 to hold.

Suppose we have a simple graph on 4 nodes: $V = (Z, X_1, X_2, Y)$. The edges are

$$Z \to X_1 \to Y$$
$$Z \to X_2 \to Y$$

So there are 2 paths from $Z$ to $Y$. The basic idea is to let $X_1, X_2$ have the same effect on $Y$, while $Z$ has opposite effects on $X_1, X_2$. Then when $Z$ is changed, the distribution of $Y$ is similar. Assuming a logistic model for each conditional probability distribution on this graph:

$$P(X_i = 1 \mid Z = z) = \sigma(\beta_i z + \alpha_i) \quad i = 1, 2$$
$$P(Y = 1 \mid X_1 = x_1, X_2 = x_2) = \sigma(\beta_0(x_1 + x_2) + \alpha_0)$$

where $\sigma(x) = 1/(1 + \exp(-x))$. It suffices to choose parameters so that $I(Z; Y) = 0$, i.e. $P(Y = 1 \mid Z = 1) = P(Y = 1 \mid Z = 0)$. We have

$$\begin{aligned}
P(Y = 1 \mid Z = 1) = &\sigma(\beta_1 + \alpha_1)\sigma(\beta_2 + \alpha_2)\sigma(2\beta_0 + \alpha_0) \\
&+ \sigma(\beta_1 + \alpha_1)\sigma(-\beta_2 - \alpha_2)\sigma(\beta_0 + \alpha_0) \\
&+ \sigma(-\beta_1 - \alpha_1)\sigma(\beta_2 + \alpha_2)\sigma(\beta_0 + \alpha_0) \\
&+ \sigma(-\beta_1 - \alpha_1)\sigma(-\beta_2 - \alpha_2)\sigma(\alpha_0)
\end{aligned}$$

---

**Algorithm 5** Backward phase of IAMB algorithm

---

**Input:** $X = (X_1, \cdots, X_d)$, $k$, $\widehat{A}_j$, $\kappa$
**Output:** Markov boundary estimate $\widehat{m}_{jk}$

1. Initialize $\widehat{m}_{jk} = \widehat{A}_j$
2. $\widehat{m}_{jk} = \widehat{m}_{jk} \setminus \big\{ X_\ell \in \widehat{m}_{jk} : \widehat{I}(X_\ell; X_k \,|\, \widehat{m}_{jk} \setminus X_\ell) < \kappa \big\}$.
3. Return $\widehat{m}_{jk}$

---

and

$$
\begin{aligned}
P(Y = 1 \,|\, Z = 0) =& \sigma(\alpha_1)\sigma(\alpha_2)\sigma(2\beta_0 + \alpha_0) \\
&+ \sigma(\alpha_1)\sigma(-\alpha_2)\sigma(\beta_0 + \alpha_0) \\
&+ \sigma(-\alpha_1)\sigma(\alpha_2)\sigma(\beta_0 + \alpha_0) \\
&+ \sigma(-\alpha_1)\sigma(-\alpha_2)\sigma(\alpha_0),
\end{aligned}
$$

so to make them equal it suffices to have

$$
\begin{aligned}
\beta_1 + \alpha_1 = \alpha_2 \\
\beta_2 + \alpha_2 = \alpha_1
\end{aligned}
$$

This implies $\beta_1 + \beta_2 = 0$, the effects from $Z$ are neutralized.

By a similar argument, this example can be generalized to $n$ nodes in the middle layer, i.e. $V = (Z, X_1, \ldots, X_n, Y)$ and $Z \to X_i \to Y$ for each $i$. In this case, it suffices to have $\sum_{i=1}^n \beta_i = 0$. This can be further generalized to $n$ paths from $Z$ to $Y$ with $m_1, \cdots, m_n$ nodes on each path,

$$
\begin{aligned}
Z &\to X_{11} \to X_{12} \to \cdots \to X_{1m_1} \to Y \\
Z &\to X_{21} \to X_{22} \to \cdots \to X_{2m_2} \to Y \\
&\cdots \\
Z &\to X_{n1} \to X_{n2} \to \cdots \to X_{nm_n} \to Y
\end{aligned}
$$

The requirement now is that there exists one node in each path

$$
\begin{aligned}
X_{1i_1} &\in \{X_{11}, X_{12}, \cdots, X_{1n_1}\} \\
X_{2i_2} &\in \{X_{21}, X_{22}, \cdots, X_{2n_2}\} \\
&\cdots \\
X_{ni_n} &\in \{X_{n1}, X_{n2}, \cdots, X_{nm_n}\}
\end{aligned}
$$

whose coefficients satisfy the equation we derived above.

*Remark* 5. Although the previous examples assume exact independence between $Y$ and $Z$ (i.e. $I(Y; Z) = 0$), this argument can be generalized as long as $I(Y; Z) < \inf_k I(Y; X_k)$.

## C.5 Relaxing Condition 2

In the previous section, we constructed examples of DAGs satisfying Condition 2 that were not poly-forests. This confirms that our results apply more broadly than Theorem 5.2 would suggest. In this section, we show that this condition can be eliminated altogether by sacrificing the sample complexity. Here we simply replace the PPS algorithm with direct estimation of $H(X_k \,|\, \widehat{A}_j)$. Then we can apply the backward phase of the IAMB algorithm [42] to infer the parents of each node in $\widehat{L}_{j+1}$ from $\widehat{A}_j$ in order to learn the whole graph. Algorithm 5 describes the backward phase of IAMB algorithm tailored for Algorithm 1, which essentially uses $\widehat{A}_j$ as a candidate set and conditional mutual information (with a parameter $\kappa$) as an independence test.

Recall that the forward phase of the IAMB algorithm is the same as the PPS procedure given in Algorithm 2. When Condition 2 fails, the estimated Markov boundary after the forward phase is no longer guaranteed to be strictly smaller than the input candidate set without further assumptions. Therefore, the forward phase does not provide a benefit in terms of the sample complexity. Denote

$$
\Delta_I = \min_j \min_{k \in L_{j+1}} \min_{\ell \in \mathrm{pa}(k)} I(X_\ell; X_k \,|\, A_j \setminus X_\ell).
$$

Condition 3 implies that $\Delta_I > 0$. Then we state the following result when Condition 2 is relaxed, recall that $w = \max_j d_j$ is the width of the DAG.

**Theorem C.7.** *Suppose $G$ satisfies the identifiability conditions in Theorem 3.1. Applying Algorithm 1, and estimate the parents of each node with Algorithm 5. If $\omega \leq \eta/2$, $\kappa \leq \Delta_I/2$, and*

$$n \gtrsim \left( \frac{2^d \sqrt{wr/d}}{\min(\kappa, \Delta/2, \omega)\sqrt{\epsilon}} \vee \frac{d^3 wr}{(\min(\kappa, \Delta/2, \omega))^2 \epsilon} \right),$$

*then $\widehat{G} = G$ with probability $1 - \epsilon$.*

The proof is deferred to Appendix H. Theorem C.7 shows for a DAG satisfying the identifiability conditions in Theorem 3.1 but without assuming Condition 2, the current analysis in Appendix E of the algorithm has exponential sample complexity in the worst-case. In fact, the sample complexity for recovering layers and then learning parents from layers using Algorithm 5 are similar. The latter simply inflates the former with $d/r$. Again, the exponential dependency comes from estimating the entropy.

### C.6 Extension to general distributions

The proofs of the theorems in this section are analogous to that of Theorem 5.1, and hence omitted.

**General discrete distributions** The sample complexity result in Theorem 5.1 can be easily extended to general discrete distributions with finite support size. The proof of Theorem 3.1 applies without change.

**Theorem C.8.** *Suppose $G$ is an arbitrary DAG with discrete variables, whose support size is bounded by $N$, satisfying the identifiability conditions in Theorem 5.1. Algorithm 3 is applied with minimax entropy estimator and thresholds in Theorem 5.1. If $M \lesssim \log d$ and sample size satisfies*

$$n \gtrsim \left( \frac{d^2 r \log^3 d \log^2 N}{(\Delta_{\omega,\kappa}^*)^2 \epsilon} \vee \frac{d^{1+\log N}}{\Delta_{\omega,\kappa}^* \log N} \sqrt{\frac{r}{\epsilon \log d}} \right),$$

*then $\widehat{G} = G$ with probability $1 - \epsilon$.*

**Continuous distributions** Theorem 3.1 also applies to continuous variables if we replace Shannon entropy with differential entropy. Differential entropy does not preserve all the properties of entropy, e.g. it is not always non-negative and it is not invariant to invertible transformations. Fortunately, differential entropy preserves the essential properties for Theorem 3.1 to hold. In particular, since continuous mutual information is still non-negative, the positiveness in Condition 3 is still reasonable to assume.

For differential entropy estimation in Algorithm 3, we can adopt the minimax estimator from Han et al. [22], which has the optimal rate over Lipschitz balls. Thus we have sample complexity result in Theorem C.9, whose proof is similar with Theorem 5.1 in Appendix E, simply replacing the estimator for entropy with the one for differential entropy and then applying the result in Han et al. [22].

**Theorem C.9.** *Suppose $G$ is an arbitrary DAG with continuous variables, whose densities are over Lipschitz balls with smoothness parameter $0 < s \leq 2$, satisfying the conditions in Theorem 5.1. Algorithm 3 is applied with differential entropy estimation, and thresholds in Theorem 5.1. If $M \lesssim \log d$ and sample size satisfies*

$$(n \log n)^{\frac{2s}{s+\log d}} \vee n \gtrsim \frac{d^2 r \log d}{(\Delta_{\omega,\kappa}^*)^2 \epsilon},$$

*then $\widehat{G} = G$ with probability $1 - \epsilon$.*

### C.7 Unfaithful example

To provide a comparison with the commonly assumed faithfulness assumption, here we construct a simple unfaithful example to illustrate how our approach does not rely on this assumption.

Consider the three-node DAG $Z \to Y$, $Z \to X \to Y$, where $Z$ is a common cause of $X$ and $Y$, and the effect $Z \to Y$ is cancelled by the path $Z \to X \to Y$. Then independence between $Z$ and $Y$ does not imply the $d$-separation between them. Now add one more node $W$ with an edge $W \to X$; See below:

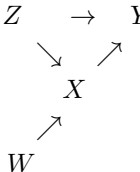

This BN is still unfaithful due to the independence between $Z$ and $Y$, but can easily be made to satisfy Condition 1. To see this, in the first layer, $W$ is used to mask $(X, Y)$ (note that $Z$ is unable to, by independence). Then in the second layer, $X$ will mask $Y$. A concrete example is following:

$$W \sim \mathcal{N}(0, \tfrac{1}{2})$$
$$Z \sim \mathcal{N}(0, 1)$$
$$X = W + \tfrac{1}{2}Z + \mathcal{N}(0, \tfrac{1}{2})$$
$$Y = X - \tfrac{1}{2}Z + \mathcal{N}(0, 1)$$

We have run experiments to show PC/GES will have SHD around 3 (i.e. very bad for this small model) but the proposed TAM algorithm perfectly recovers the DAG, as expected.

Furthermore, this simple example can be embedded into any DAG satisfying our identifiability condition by treating some sink node as $W$. And we may conjecture that, when unfaithfulness happens in a form of path-cancellation, if we have some other nodes in the "ancestral" layer, it is able to help identify the DAG as long as the "uncertainty" relation in (C1) is satisfied.

## D Proof of Theorem 3.1

Since Theorem 3.1 is a special case of Theorem C.4 (i.e. $\mathrm{an}_j^i(k) = \emptyset$), it suffices to prove the latter result.

*Proof of Theorem C.4.* The proof illustrates the mechanism of Algorithm 4 to identify DAGs satisfying conditions in Theorem C.4. If we have identified $A_j = \cup_{t=0}^{j} L_t$, our goal is to distinguish $L_{j+1}$ from $V \setminus A_{j+1}$. If it is possible, then by induction on $j$ we can complete the proof.

Denote $H(X_k \mid A_j) = h_{jk}$ for all $k \in V \setminus A_j$. Sort $h_{jk}$ in ascending order, denote this order by $\tau^{(0)}$, where $\tau^{(0)}$ is a permutation of size $|V \setminus A_j|$. Any node $X_k \in V \setminus A_{j+1}$ must have at least one ancestor in $L_{j+1}$. Meanwhile, there exists one important ancestor $X_i \in \mathrm{an}_j(k)$ such that $h_{ji} < h_{jk}$ by (U1). This implies $\tau_i^{(0)} < \tau_k^{(0)}$ and therefore, $\tau_1^{(0)}$ must be from $L_{j+1}$.

We proceed by considering two operations on the nodes in $V \setminus A_j$: Testing and Masking. Specifically, we maintain two sequences of set of nodes $\widetilde{L}^{(t)}$ and $\widetilde{S}^{(t)}$, which is indexed by number of testing operations we have conducted (starts with $t = 0$ and the same with the superscript of $\tau^{(t)}$). Initialize $\widetilde{L}^{(0)} = \widetilde{S}^{(0)} = \emptyset$, and put nodes being tested or masked into $\widetilde{L}^{(t-1)}$ or $\widetilde{S}^{(t-1)}$ to get $\widetilde{L}^{(t)}$ or $\widetilde{S}^{(t)}$ respectively.

Then we for each $t = 1, 2, \cdots$, update

$$\widetilde{L}^{(t)} = \widetilde{L}^{(t-1)} \cup \{\tau_1^{(t-1)}\}$$

If there are ties, include all the tied nodes into $\widetilde{L}^{(t)}$ and follow the same argument. For $k \in \tau_{[2:]}^{(t-1)}$ compute the mutual information (a.k.a. entropy reduction)

$$I(X_k; \widetilde{L}^{(t)} \mid A_j) = h_{jk} - H(X_k \mid A_j, \widetilde{L}^{(t)})$$

and mask the nodes with positive mutual information, namely update the masked set

$$\widetilde{S}^{(t)} = \widetilde{S}^{(t-1)} \cup \{k : I(X_k; \widetilde{L}^{(t)} \mid A_j) > 0\}.$$

Then remove the nodes that have been conditioned or masked from $\tau^{(t-1)}$ to get

$$\tau^{(t)} = \tau^{(t-1)} \setminus (\widetilde{S}^{(t)} \cup \widetilde{L}^{(t)})$$

The following crucial properties of an important ancestor $X_i$ with $\mathrm{an}_j^i(k)$ of $X_k$ as easy to check:

1. $X_i \perp\!\!\!\perp (L_{j+1} \setminus X_i) \,|\, A_j$, and thus $I(X_i; L_{j+1} \setminus X_i \,|\, A_j) = 0$, so is $\mathrm{an}_j^i(k)$;

2. For $X_\ell \in \mathrm{an}_j^i(k)$, $\tau_\ell^{(t)} < \tau_i^{(t)}$ if $X_i$ and $X_\ell$ are in $\tau^{(t)}$;

3. $(X_i, \mathrm{an}_j^i(k))$ will not be masked and put into $\widetilde{S}^{(t)}$ for all $t$;

4. If $X_k \in \tau^{(t)}$, then $X_i \in \tau^{(t)}$ and $\tau_i^{(t)} < \tau_k^{(t)}$.

1) is due to $(X_i, \mathrm{an}_j^i(k)) \subseteq L_{j+1}$ and the definition of layer decomposition. 2) is by the definition of $\mathrm{an}_j^i(k)$, thus $\tau_\ell^{(0)} < \tau_i^{(0)}$. And $\tau^{(t)}$ is a subset of $\tau^{(0)}$, hence the order is preserved. 3) is by 1) otherwise the mutual information is positive. For 4), if $X_i \notin \tau^{(t)}$, then $X_i$ has been included into $\widehat{L}^{(t)}$, so is $\mathrm{an}_j^i(k)$ by 2). Then $X_k$ should have been masked before $t$ since

$$I(X_k; \widetilde{L}^{(t)} \,|\, A_j) \geq I(X_k; (X_i, \mathrm{an}_j^i(k)) \,|\, A_j) > 0.$$

$\tau_i^{(t)} < \tau_k^{(t)}$ is due to the same reason as 2).

Then we conclude that $\tau_1^{(t)}$ must be from $L_{j+1}$ by 4). By continuing this procedure, all the nodes are eventually tested or masked after say $t^*$ steps, i.e., $V \setminus A_j = \widetilde{L}^{(t^*)} \cup \widetilde{S}^{(t^*)}$. Since nodes in $\widetilde{L}^{(t^*)}$ are composed of $\tau_1^{(t)}$ for $t = 1, 2, \cdots$, thus $\widetilde{L}^{(t^*)} \subseteq L_{j+1}$. Then we further claim for any $\ell \in \widetilde{S}^{(t^*)}$, $X_\ell \notin L_{j+1}$. Suppose $X_\ell$ is included at step $t$, otherwise we will have

$$0 < I(X_\ell; \widetilde{L}^{(t)} \,|\, A_j) \leq I(X_\ell; L_{j+1} \,|\, A_j) = 0$$

The last equality is due to the mutual independence of nodes in $L_{j+1}$ given $A_j$. Therefore, the final $\widetilde{L}^{(t^*)}$ is exactly $L_{j+1}$. $\qquad\square$

# E  Proof of Theorem 5.1

We prove the theorem in two steps. After establishing some preliminary bounds, we prove Proposition 4.1, which gives the sample complexity for using PPS to recover Markov boundaries and estimate the conditional entropy under Condition 2. Then we establish the sample complexity of the layer-wise learning framework for DAGs satisfying identifiability conditions in Theorem 3.1.

## E.1  Preliminary bounds

To derive the sample complexity of Algorithm 3, we borrow the minimax estimator of entropy from Wu and Yang [47]. Denote this estimator by $\widehat{H}(X)$.

**Lemma E.1.** *For discrete random variables* $X_1, \cdots, X_d$ *with distribution* $P_1, \cdots, P_d$ *and support size* $K_1, \cdots, K_d$, *let* $K = \max_k K_k$. *If* $n \gtrsim \frac{K}{\log K}$, *then*

$$\sup_k P\left[\left|H(X_k) - \widehat{H}(X_k)\right| \geq t\right] \lesssim \left[\left(\frac{K}{n \log K}\right)^2 + \frac{\log^2 K}{n}\right]/t^2$$

*Proof.* First we pad all the random variables such that they have the same support size $K$ by assigning the extra support with zero probability. Then for all $X_k$

$$P\left[\left|H(X_k) - \widehat{H}(X_k)\right| \geq t\right] \leq \frac{\mathbb{E}(H(X_k) - \widehat{H}(X_k))^2}{t^2}$$

Thus, by Proposition 4 in Wu and Yang [47],

$$\sup_k P\left[\left|H(X_k) - \widehat{H}(X_k)\right| \geq t\right] \leq \frac{\sup_k \mathbb{E}(H(X_k) - \widehat{H}(X_k))^2}{t^2}$$

$$\lesssim \frac{1}{t^2}\left[\left(\frac{K}{n\log K}\right)^2 + \frac{\log^2 K}{n}\right].$$

□

We next establish some preliminary uniform bounds on the estimation error of conditional entropy and mutual information using Lemma E.1. Suppose we estimate the conditional entropy $H(X_k \mid A)$ for any $k \in V$ and its some ancestral set $A$ by

$$\widehat{H}(X_k \mid A) = \widehat{H}(X_k, A) - \widehat{H}(A). \tag{7}$$

Then because we are dealing with binary variables, by Lemma E.1

$$\sup_{k,A} P\left[\left|\widehat{H}(X_k \mid A) - H(X_k \mid A)\right| \geq t\right]$$

$$\leq \sup_{k,A} \frac{1}{t^2}\left(\widehat{H}(X_k \mid A) - H(X_k \mid A)\right)^2$$

$$\leq \sup_{k,A} \frac{1}{t^2}\left\{2\left(\widehat{H}(X_k, A) - H(X_k, A)\right)^2 + 2\left(\widehat{H}(A) - H(A)\right)^2\right\}$$

$$\lesssim \frac{\delta^2_{|A|+1}}{t^2} \tag{8}$$

where

$$\delta^2_p \asymp \left[\left(\frac{2^p}{np}\right)^2 + \frac{p^2}{n}\right]. \tag{9}$$

Note that $\delta^2_p$ is an increasing function of $p$. When $A = \emptyset$, and the $|A| = 0$. Thus $\delta^2_{|A|+1}$ reduces to estimating entropy of a binary random variable, which is of parametric rate $1/n$. Similarly, if we try to estimate the conditional mutual information by using sample version of the identity

$$I(X_k; X_\ell \mid A) = I(X_k; (X_\ell, A)) - I(X_k; A)$$
$$= H(X_k) + H(X_\ell, A) - H(X_k, X_\ell, A)$$
$$- H(X_k) - H(A) + H(X_k, A)$$
$$= H(X_\ell, A) - H(X_k, X_\ell, A) - H(A) + H(X_k, A)$$

The estimation error is again dominated by the second term, which has the largest support size:

$$\sup_{k,\ell,A} P\left[\left|\widehat{I}(X_k; X_\ell \mid A) - I(X_k; X_\ell \mid A)\right| \geq t\right] \lesssim \frac{\delta^2_{|A|+2}}{t^2} \tag{10}$$

The factor of $+2$ is not important when size $|A|$ is large, thus for simplicity we absorb it into the constant before them. We will present the estimation error bound for $H(X_\ell \mid A)$ and $I(X_k; X_\ell \mid A)$ by $C\delta^2_{|A|}/t^2$ for some constant $C$.

## E.2  Proof of Proposition 4.1

In this section, we prove the Proposition 4.1. Recall that $M_{Ak} := |\operatorname{MB}(k; A)|$.

*Proof of Proposition 4.1.* For any node $k \in V$ and its ancestor set $A$, and node $\ell \in A$, (10) implies

$$\sup_{A' \subseteq A, |A'| \leq M_{Ak}} P\left[\left|\widehat{I}(X_k; X_\ell \mid A') - I(X_k; X_\ell \mid A')\right| \geq t\right] \lesssim \frac{\delta^2_{M_{Ak}}}{t^2}.$$

For the first step, with probability $1 - |A|\delta^2_{M_{Ak}}/t^2$ we have for all $X_\ell \in A$

$$\left|\widehat{I}(X_k; X_\ell) - I(X_k; X_\ell)\right| < t$$

Thus for all $X_{\ell'} \in A \setminus m_{Ak}$,

$$\widehat{I}(X_k; X_{\ell^*}) - \widehat{I}(X_k; X_{\ell'}) > \tilde{\Delta}_{Ak} - 2t$$

where $\ell^* = \arg\max_{i:X_i \in m_{Ak}} I(X_k; X_i)$. So we only need $t < \tilde{\Delta}_{Ak}/2$ to ensure we include a node in $m_{Ak}$ rather than in $A \setminus m_{Ak}$. Following the same argument, when we have found a proper subset $m \subsetneq m_{Ak}$, with probability $1 - (|A| - |m|)\,\delta^2_{M_{Ak}}/t^2$ we have for all $X_\ell \in A \setminus m$

$$\left|\widehat{I}(X_k; X_\ell \,|\, m) - I(X_k; X_\ell \,|\, m)\right| < t.$$

Thus for all $X_{\ell'} \in A \setminus m_{Ak}$

$$\widehat{I}(X_k; X_{\ell^*} \,|\, m) - \widehat{I}(X_k; X_{\ell'} \,|\, m) > \tilde{\Delta}_{Ak} - 2t$$

where $\ell^* = \arg\max_{i:X_i \in m_{Ak} \setminus m} I(X_k; X_i \,|\, m)$. So we only need $t \leq \tilde{\Delta}_{Ak}/2$ to ensure we do not include any nodes from $A \setminus m_{Ak}$. At the same time,

$$\widehat{I}(X_k; X_{\ell^*} \,|\, m) > I(X_k; X_{\ell^*} \,|\, m) - t \geq 2\xi_{Ak} - t$$

To avoid triggering the threshold, we need

$$\widehat{I}(X_k; X_{\ell^*} \,|\, m) > 2\xi_{Ak} - t \geq \kappa.$$

So $t \leq \kappa$ will do the job. After at most $M_{Ak}$ steps, we have recovered $m_{Ak}$, then requiring $t \leq \kappa$ will trigger the stopping criterion. Since for any $X_{\ell'} \in A \setminus m_{Ak}$,

$$\left|\widehat{I}(X_k; X_{\ell'} \,|\, m_{Ak}) - I(X_k; X_{\ell'} \,|\, m_{Ak})\right| = \widehat{I}(X_k; X_{\ell'} \,|\, m_{Ak}) < t \leq \kappa.$$

Thus in conclusion, we can recover $m_{Ak}$ for $X_k$ with probability

$$P(\widehat{m}_{Ak} = m_{Ak}) \geq \prod_{i=0}^{M_{Ak}} \left(1 - (|A| - |i|)\frac{\delta^2_{M_{Ak}}}{t^2}\right)$$

$$\geq 1 - \left((M_{Ak} + 1)|A| - \frac{M_{Ak}(M_{Ak} + 1)}{2}\right)\frac{\delta^2_{M_{Ak}}}{t^2}$$

$$\geq 1 - (M_{Ak} + 1)|A|\frac{\delta^2_{M_{Ak}}}{t^2}.$$

Furthermore, by combining this with (8), we can estimate the conditional entropy as

$$P\left[\left|\widehat{H}(X_k \,|\, A) - H(X_k \,|\, A)\right| < t', \widehat{m}_{Ak} = m_{Ak}\right]$$

$$= P(\widehat{m}_{Ak} = m_{Ak})P\left[\left|\widehat{H}(X_k \,|\, A) - H(X_k \,|\, A)\right| < t' \,|\, \widehat{m}_{Ak} = m_{Ak}\right]$$

$$= P(\widehat{m}_{Ak} = m_{Ak})P\left[\left|\widehat{H}(X_k \,|\, m_{Ak}) - H(X_k \,|\, m_{Ak})\right| < t'\right]$$

$$\geq \left(1 - (M_{Ak} + 1)|A|\frac{\delta^2_{M_{Ak}}}{t^2}\right)\left(1 - \frac{\delta^2_{M_{Ak}}}{t'^2}\right)$$

$$\geq \left(1 - (M_{Ak} + 1)|A|\frac{\delta^2_{M_{Ak}}}{t^2} - \frac{\delta^2_{M_{Ak}}}{t'^2}\right)$$

Let $t' = t \leq \min(\kappa, \tilde{\Delta}_{Ak}/2)$, then we have

$$P\left[\left|\widehat{H}(X_k \,|\, A) - H(X_k \,|\, A)\right| < t, \widehat{m}_{Ak} = m_{Ak}\right] \geq 1 - (M_{Ak} + 2)|A|\frac{\delta^2_{M_{Ak}}}{t^2}$$

The argument above holds for all $X_k \in V$ and its ancestor set $A$, which completes the proof. $\quad\square$

### E.3 Proof of Theorem 5.1

Now we are ready to prove the main theorem on the sample complexity of Algorithm 3.

*Proof of Theorem 5.1.* Define events by $\mathcal{E}_0 = \emptyset$ and

$$\mathcal{E}_j = \left\{ \widehat{L}_j = L_j \text{ and } \widehat{m}_{(j-1)k} = m_{(j-1)k} \text{ for } k \in L_j \right\}$$

for $j = 1, \ldots, r$. Then

$$P(\widehat{G} = G) = \prod_{j=1}^{r} P(\mathcal{E}_j \mid \mathcal{E}_{j-1}).$$

For the first step, with probability $1 - d\delta_0^2/t^2$ we have for all $k = 1, \cdots, d$

$$|H(X_k) - \widehat{H}(X_k)| < t$$

which implies for $k \notin L_1$ and its corresponding $X_i$

$$\widehat{H}(X_k) - \widehat{H}(X_i) > \Delta - 2t$$

With $t \leq \Delta/2$, we have $X_i$ comes before $X_k$ in the order $\widehat{\tau}$ of estimated marginal entropies. Conducting the TAM step, for each $X_i$ comes forst in the ordering $\widehat{\tau}$, we estimate the mutual information $I(X_i; X_k)$ for all remaining nodes $X_k$ in $\widehat{\tau}$. With probability at least $1 - d\delta_2^2/t^2$, we have for all remaining $k$

$$|I(X_i; X_k) - \widehat{I}(X_i; X_k)| < t$$

Therefore,

$$\begin{cases} I(X_i; X_k) < 2t & X_i \text{ is not the ancestor of } X_k \\ I(X_i; X_k) > \eta - 2t & X_i \text{ is the ancestor of } X_k \end{cases}$$

With $\omega := t \leq \eta/2$, $X_k$ would be masked. There are at most $d_1 := |L_1|$ many $X_i$'s thus there is at most $d_1 \times d$ mutual information need to be estimated correctly. So with probability at least $1 - d_1 d\delta_2^2/t^2$, the TAM step succeeds recovering $L_1$. And then

$$P(\mathcal{E}_1) \geq 1 - \frac{d\delta_0^2}{\Delta^2/4} - \frac{d_1 d\delta_2^2}{\omega^2}$$

Following the same argument, after $j$ loops, given the layers in $A_j$ are correctly identified, we invoke Proposition 4.1 with $A = A_j$ to have for all $k \in V \setminus A_j$, with probability at least

$$1 - \left( \sum_{s=j+1}^{r} d_s \right) \left( 2M_j \sum_{s=1}^{j} d_s \frac{\delta_{M_j}^2}{t^2} \right),$$

using PPS procedure to estimate the conditional entropeis and Markov boundaries gives

$$|\widehat{H}(X_k \mid A_j) - H(X_k \mid A_j)| < t \quad \text{and} \quad \widehat{m}_{jk} = m_{jk}.$$

which implies for $k \notin L_{j+1}$ and its corresponding $X_i$,

$$\widehat{H}(X_k \mid A_j) - \widehat{H}(X_i \mid A_j) > \Delta - 2t$$

So $t \leq \Delta/2$ amounts to $X_i$ coming before $X_k$ in the order $\widehat{\tau}$ of estimated conditional entropies of remaining nodes. Note that $t$ also needs to satisfy $t \leq \min_k(\kappa, \widetilde{\Delta}_{jk}/2)$. Conducting TAM step, estimate the conditional mutual information using the identity

$$I(X_k; X_i \mid A_j) = H(X_k \mid A_j) - H(X_k \mid X_i, A_j)$$
$$= H(X_k \mid m_{jk}) - H(X_k \mid X_i, m_{jk})$$

Invoking (8), since $m_{jk}$ is already identified, with probability at least $1 - d\delta_{M_j}^2/t^2$, we have for all remaining $k$

$$|I(X_i; X_k) - \widehat{I}(X_i; X_k)| < t$$

Therefore,

$$\begin{cases} I(X_i; X_k) < 2t & X_i \text{ is not the ancestor of } X_k \\ I(X_i; X_k) > \eta - 2t & X_i \text{ is the ancestor of } X_k \end{cases}$$

With $\omega := t \leq \eta/2$, $X_k$ would be masked while other nodes remains unmasked. There are at most $d_{j+1} := |L_{j+1}|$ manny $X_i$'s thus there is at most $d_{j+1} \times d$ conditional mutual information need to be estimated correctly. So with probability at least $1 - d_{j+1} d \delta_{M_j}^2 / t^2$, the TAM step succeeds recovering $L_{j+1}$. Combine the PPS step and TAM step together,

$$P(\mathcal{E}_{j+1} \,|\, \mathcal{E}_j) \geq 1 - \frac{d_{j+1} d \delta_{M_j}^2}{\omega^2} - \frac{2 M_j d^2 \delta_{M_j}^2}{\left( \min_k(\Delta/2, \kappa, \widetilde{\Delta}_{jk}/2) \right)^2}$$

In conclusion, we have

$$P(\widehat{G} = G) = \prod_{j=0}^{r-1} P(\mathcal{E}_{j+1} \,|\, \mathcal{E}_j)$$

$$\geq \left( 1 - \frac{d \delta_0^2}{\Delta^2/4} - \frac{d_1 d \delta_2^2}{\omega^2} \right) \prod_{j=1}^{r-1} \left( 1 - \frac{d_{j+1} d \delta_{M_j}^2}{\omega^2} - \frac{2 M_j d^2 \delta_{M_j}^2}{\left( \min_k(\Delta/2, \kappa, \widetilde{\Delta}_{jk}/2) \right)^2} \right)$$

$$\geq 1 - \left( 1 - \frac{d \delta_0^2}{\Delta^2/4} - \frac{d_1 d \delta_2^2}{\omega^2} \right) - \sum_{j=1}^{r-1} \left( \frac{d_{j+1} d \delta_{M_j}^2}{\omega^2} - \frac{2 M_j d^2 \delta_{M_j}^2}{\left( \min_k(\Delta/2, \kappa, \widetilde{\Delta}_{jk}/2) \right)^2} \right)$$

$$\geq 1 - \frac{4 M r d^2}{\left( \min_{jk}(\Delta/2, \omega, \kappa, \widetilde{\Delta}_{jk}/2) \right)^2} \delta_M^2$$

where $\omega \leq \eta/2$ and $\kappa \leq \min_{jk} \xi_{jk}$. Since $M \lesssim \log d$, $\delta_M^2$ (cf. (9)) is dominated by its second term. Requiring $P(\widehat{G} = G) > 1 - \epsilon$, we have the final result

$$n \gtrsim \frac{d^2 r \log^3 d}{\left( \min_{jk}(\Delta/2, \omega, \kappa, \widetilde{\Delta}_{jk}/2) \right)^2 \epsilon}.$$

This completes the proof.

$\square$

### E.4 Tuning parameters

In practice, the quantities $\eta, \xi_{jk}$ needed in Theorem 5.1 may not be known. Theorem E.2 below remedies this by prescribing data-dependent choices for $\omega$ and $\kappa$:

**Theorem E.2.** *Suppose the conditions in Theorem 5.1 are satisfied with the stronger sample size requirement*

$$n \gtrsim \frac{d^3 \log^3 d}{\epsilon^2 \wedge (\Delta_{\eta/2, \xi_{jk}}^*)^4},$$

*where $\Delta_{\eta/2, \xi_{jk}}^*$ is defined in Theorem 5.1 with $\eta/2$ and $\xi_{jk}$ plugged in. By choosing*

$$\omega = \kappa \asymp (d^3 \log d)^{1/4} \left[ \left( \frac{d}{n \log d} \right)^2 + \frac{\log d}{\sqrt{n}} \right]^{1/4}$$

*in Algorithm 3, we have $\widehat{G} = G$ with probability $1 - \epsilon$.*

*Proof.* In the proof of Proposition 4.1, for $X_k \in L_{j+1}$, we need $t$ to be small enough when doing estimation such that

$$0 \leq \tilde{\Delta}_{jk} - 2t \quad \text{and} \quad t \leq \kappa \leq 2\xi_{jk} - t.$$

Similarly, in the proof of Theorem 5.1 we need

$$\omega = t \leq \eta/2.$$

Though the $t$'s here are different, we can take the minimum of them, then it suffices to take

$$\omega = \kappa = t = (d^3 M)^{1/4} \delta_M^{1/2}$$

and require

$$(d^3 M)^{1/4} \delta_M^{1/2} \leq \min_{jk}(\Delta/2, \eta/2, \xi_{jk}, \tilde{\Delta}_{jk}/2).$$

Then we have

$$
\begin{aligned}
P(\widehat{G} = G) &\geq 1 - \frac{4d^2 r M \delta_M^2}{\left(\min_{jk}(\Delta/2, \omega, \kappa, \tilde{\Delta}_{jk}/2)\right)^2} \\
&\geq 1 - \frac{4d^3 M \delta_M^2}{\left(\min_{jk}(\Delta/2, \omega, \kappa, \tilde{\Delta}_{jk}/2)\right)^2} \\
&\geq 1 - 4d\sqrt{dM}\delta_M.
\end{aligned}
$$

Plugging in the exact form of $\delta_M$ with $M \lesssim \log d$, the final sample complexity is as desired. $\quad\square$

# F   Proof of Theorem 5.2

First, we need a lemma similar to Lemma C.1:

**Lemma F.1.** $I(X_k; m_{jk} \mid A) > 0$ *for any subset $A \subset A_j$ such that $m_{jk} \setminus A \neq \emptyset$.*

The proof follows the one for Lemma C.1 in Appendix C.1. To see this, we can simply replace $\mathrm{pa}(k)$ in the arguments with $m_{jk}$, and replace the minimality of $G$ with minimality of $m_{jk}$. We will also need the following lemma:

**Lemma F.2.** *For any $X_c \in m_{jk}$, there is at least one directed path from $X_c$ to $X_k$.*

*Proof.* If there is no path between $X_c$ and $X_k$, $X_c \perp\!\!\!\perp X_k$, $X_c$ cannot be in $m_{jk}$. If $X_c$ and $X_k$ are only connected by undirected paths, $X_c$ would not be in ancestors or descendants of $X_k$. If some paths are through some descendants (children) of $X_k$, the descendants would serve as colliders to block the paths, so the effective paths must be through the ancestors of $X_k$.

For any undirected path connects them through some ancestor, say $X_a$, if the edge from $X_c$ is not pointing to $X_a$, $X_a$ would serve as a common cause. If it is pointing to $X_a$, we move $X_a$ one node toward $X_c$, until there is a common cause, otherwise $X_c$ is connected to $X_k$ with a directed path. When we find the common cause, if $X_a$ is an ancestor of $X_c$, then it is in $A_j$ and block the path if conditioned on. If $X_a$ is not an ancestor of $X_c$, then there must be a change of edge direction, so there exits a collider on the path between $X_a$ and $X_c$. If the collider is not in $A_j$, it will block the path; if it is in $A_j$, $X_a$ will also be in $A_j$, so the path is blocked when $A_j$ is conditioned on. As a result of all the cases above,

$$P(X_k \mid A_j) = P(X_k \mid A_j \setminus X_c)$$

so $X_c$ cannot be in $m_{jk}$. $\quad\square$

Finally we prove the theorem:

*Proof of Theorem 5.2.* Condition 2 is constructive for PPS procedure. Hence we prove it by showing that nodes not belong to the desired Markov boundary will not be chosen when conducting PPS on poly-forest. Suppose we have identified $j$-th layer, for any $X_k \in V \setminus A_j$, we try to find $m_{jk}$. For any node $X_\ell \in A_j \setminus m_{jk}$ we should not consider, if $X_\ell$ is disconnected to $X_k$, which means there is no path connecting them together, then $X_k \perp\!\!\!\perp X_\ell \,|\, m$, $I(X_k; X_\ell \,|\, m) = 0$ for any $m \subsetneq m_{jk}$, we do not need to worry about it.

If $X_\ell$ is connected to $X_k$ through a child of $X_k$, then this must be an undirected path. Since if this is a directed path, it will be from $X_k$ to $X_\ell$, which is contradicted to $X_\ell \in A_j$. Moreover for this undirected path, since the edge is from $X_k$ to its child at first, there will be a change of direction, which serves as a collider blocking this and the only path connecting $X_k$ and $X_\ell$, thereafter $X_k$ and $X_\ell$ are $d$-separated by any subset $m \subsetneq m_{jk}$ and $X_k \perp\!\!\!\perp X_\ell \,|\, m$, $I(X_k; X_\ell \,|\, m) = 0$, we do not need to worry about it.

If $X_\ell$ is connected to $X_k$ through one parent $X_t$ or $X_\ell$ is the parent, we can divide it into two situations, whether there is $X_c \in m_{jk}$ on the path or not. If there is one $X_c$ on the path, since $X_c$ has exactly one directed path leading to $X_k$, this will be part of the path connecting $X_\ell$ and $X_k$. For two edges of $X_c$ on this path, $X_c$ has an edge out on the path toward $X_k$. For the other edge, no matter the direction is, either in or out, $X_c$ will block the path if conditioned on and $d$-separates $X_k$ and $X_\ell$. Then we have for any $m \subsetneq m_{jk}$

$$X_k \perp\!\!\!\perp X_\ell \,|\, X_c, m$$

Using this property and decomposition

$$\begin{aligned} I(X_k; X_c, X_\ell \,|\, m) &= I(X_k; X_c \,|\, X_\ell, m) + I(X_k; X_\ell \,|\, m) \\ &= I(X_k; X_\ell \,|\, X_c, m) + I(X_k; X_c \,|\, m) = I(X_k; X_c \,|\, m) \end{aligned}$$

So

$$I(X_k; X_c \,|\, m) = I(X_k; X_c \,|\, X_\ell, m) + I(X_k; X_\ell \,|\, m) > I(X_k; X_\ell \,|\, m)$$

For the last inequality, we trigger Lemma F.1. So we will not select $X_\ell$ at any step of PPS procedure.

If there is no $X_c$ on the path, then for any $X_c \in m_{jk}$, since $X_c$ is connected to $X_k$ through a directed path, and $X_\ell$ is connected to $X_k$ through one parent $X_t$ or $X_\ell$ is the parent, then $X_c$ and $X_\ell$ is connected by and only by the combination of two paths $X_c \to \cdots \to X_k$ and $X_\ell - \cdots - X_t \to X_k$. If these two paths converge at $X_k$, $X_k$ will serve as a collider to block this path. If they converge at some node before $X_k$, denoted as $X_u$, which is not in $m_{jk}$. $X_u$ cannot be $X_\ell$ otherwise it will $d$-separate $X_k$ and $X_c$ such that $X_c \notin m_{jk}$. Since the path $X_c \to \cdots \to X_k$ is directed, the edge on this path from $X_c$ is pointing to $X_u$. If the edge from $X_\ell$ is also pointing to $X_u$, $X_u$ will serve as a collider to block this path between $X_\ell$ and $X_c$. So $X_c$ and $X_\ell$ are $d$-separated by empty set in these two cases, furthermore $m_{jk} \perp\!\!\!\perp X_\ell$. Since $X_\ell \perp\!\!\!\perp X_k \,|\, m_{jk}$. Therefore we have

$$\begin{aligned} P(X_\ell, X_k, m_{jk}) &= P(X_\ell, X_k \,|\, m_{jk})P(m_{jk}) \\ &= P(X_\ell \,|\, m_{jk})P(X_k \,|\, m_{jk})P(m_{jk}) \\ &= P(X_\ell)P(X_k, m_{jk}) \end{aligned}$$

Hence $X_\ell \perp\!\!\!\perp (X_k, m_{jk})$ and for any $m \subsetneq m_{jk}$

$$\begin{aligned} 0 &= I(X_\ell; (X_k, m_{jk})) \\ &= I(X_\ell; X_k \,|\, m) + I(X_\ell; m) + I(X_\ell; m_{jk} \setminus m \,|\, X_k, m) \end{aligned}$$

By non-negativity of conditional mutual information, we have $I(X_\ell; X_k \,|\, m) = 0$. So we don not need to worry about it either.

If the edge from $X_\ell$ is not pointing to $X_u$ but on the opposite, $X_\ell$ and $X_c$ may not be independent. However, for any other node $X_{c'} \in m_{jk} \setminus X_c$, if $X_\ell$ is connected with $X_{c'}$ through $X_k$, $X_k$ serves as a collider to block the path. If $X_\ell$ is connected with $X_{c'}$ through $X_c$, conditioning on $X_c$ will block the path. Therefore, $X_c$ $d$-separates $X_\ell$ and $X_{c'}$ so $X_\ell \perp\!\!\!\perp (m_{jk} \setminus X_c) \,|\, X_c$. As a result,

$$\begin{aligned} P(X_\ell, m_{jk} \setminus X_c \,|\, X_c) &= P(X_\ell \,|\, X_c)P(m_{jk} \setminus X_c \,|\, X_c) \\ P(X_\ell \,|\, X_c) &= P(X_\ell \,|\, m_{jk}) \end{aligned}$$

Thus

$$\begin{aligned} P(X_\ell, X_k, m_{jk}) &= P(X_\ell \,|\, m_{jk})P(X_k \,|\, m_{jk})P(m_{jk}) \\ &= P(X_\ell \,|\, X_c)P(X_k, m_{jk}) \end{aligned}$$

Meanwhile,
$$P(X_\ell, X_k, m_{jk}) = P(X_\ell \mid X_k, m_{jk})P(X_k, m_{jk})$$

Hence

$$P(X_\ell \mid X_k, m_{jk}) = P(X_\ell \mid X_c) \quad X_\ell \perp\!\!\!\perp (X_k, m_{jk} \setminus X_c) \mid X_c \quad I(X_\ell; (X_k, m_{jk} \setminus X_c) \mid X_c) = 0$$

By the same decomposition of mutual information, we can have for any subset $m$, $I(X_\ell; X_k \mid X_c, m) = 0$, thereafter we will not select $X_\ell$ at any step of PPS procedure. $\qquad\square$

## G  Condition (C2) and poly-forests

Condition (C2) is a nondegeneracy condition on the distribution $P$ that may be violated, for example, when there is path cancellation. In this appendix, we show that a sufficient (but not necessary) condition for Condition (C2) is that there exists at most one directed path between any two nodes in the graph, in other words, when path cancellation is impossible.

**Lemma G.1.** *If $G$ is a poly-forest, then Condition (C2) holds.*

*Proof.* Let's drop the subscript and use the notation $X, Z$ for $X_i, X_k$. For poly-forest, there exists and only exists one direct path from ancestor $X$ to descendant $Z$. With loss of generality, let $A_j = \emptyset$, since the Markov property of the subgraph $G[V \setminus A_j]$ does not change. If $X \in \mathrm{pa}(Z)$, by minimality, all edges are effective, thus $I(X; Z) > 0$.

If the directed path is formed by three nodes $X \to Y \to Z$. Suppose the contrary, $X \perp\!\!\!\perp Z$, we have following:

$$
\begin{aligned}
P(X)P(Z) &= P(X, Z) \\
&= \sum_{y \in \{0,1\}} P(X, Y = y, Z) \\
&= P(X) \sum_{y \in \{0,1\}} P(Z \mid Y = y)P(Y = y \mid X)
\end{aligned}
$$

The first equality is by independence, the third one is by Markov property. Then by positivety of the probability, we have

$$
\sum_{y \in \{0,1\}} P(Z \mid Y = y)P(Y = y \mid X) = P(Z)
$$

$$
= \sum_{y \in \{0,1\}} P(Z \mid Y = y)P(Y = y)
$$

After rearrangement,

$$
\sum_{y \in \{0,1\}} P(Z \mid Y = y)\Big[P(Y = y \mid X) - P(Y = y)\Big] = 0
$$

Since $X \not\perp\!\!\!\perp Y$, thus $P(Y = y \mid X) \neq P(Y = y)$, therefore,

$$
\frac{P(Z \mid Y = 1)}{P(Z \mid Y = 0)} = -\frac{P(Y = 0 \mid X) - P(Y = 0)}{P(Y = 1 \mid X) - P(Y = 1)} = -\frac{1 - P(Y = 1 \mid X) - 1 + P(Y = 1)}{P(Y = 1 \mid X) - P(Y = 1)} = 1
$$

Thus $P(Z \mid Y = 1) = P(Z \mid Y = 0)$ contradicts that $Y \not\perp\!\!\!\perp Z$.

More generally, let the directed path be formed by $X \to Y_1 \to \cdots \to Y_p \to Z$. Suppose $X \perp\!\!\!\perp Z$, we have:

$$
\begin{aligned}
P(X)P(Z) &= P(X, Z) \\
&= \sum_{y_1 \in \{0,1\}} \cdots \sum_{y_d \in \{0,1\}} P(X, Y_1 = y_1, \cdots, Y_d = y_d, Z) \\
&= P(X) \sum_{y_d \in \{0,1\}} \sum_{y_1 \in \{0,1\}} P(Z \mid Y_d = y_d)P(Y_d = y_d \mid Y_1 = y_1)P(Y_1 = y_1 \mid X)
\end{aligned}
$$

After rearrangement,

$$\frac{P(Z \mid Y_d = 1)}{P(Z \mid Y_d = 0)} = -\frac{\sum_{y_1 \in \{0,1\}}[P(Y_1 = y_1 \mid X) - P(Y_1 = y_1)]P(Y_d = 1 \mid Y_1 = y_1)}{\sum_{y_1 \in \{0,1\}}[P(Y_1 = y_1 \mid X) - P(Y_1 = y_1)]P(Y_d = 0 \mid Y_1 = y_1)}$$

Look at the numerator

$$\sum_{y_1 \in \{0,1\}} \left[ P(Y_1 = y_1 \mid X) - P(Y_1 = y_1) \right] P(Y_d = 1 \mid Y_1 = y_1)$$

$$= \sum_{y_1 \in \{0,1\}} \left[ P(Y_1 = y_1 \mid X) - P(Y_1 = y_1) \right] \left[ 1 - P(Y_d = 0 \mid Y_1 = y_1) \right]$$

$$= -\sum_{y_1 \in \{0,1\}} \left[ P(Y_1 = y_1 \mid X) - P(Y_1 = y_1) \right] P(Y_d = 0 \mid Y_1 = y_1)$$

$$+ \sum_{y_1 \in \{0,1\}} P(Y_1 = y_1 \mid X) - \sum_{y_1 \in \{0,1\}} P(Y_1 = y_1)$$

$$= -\sum_{y_1 \in \{0,1\}} \left[ P(Y_1 = y_1 \mid X) - P(Y_1 = y_1) \right] P(Y_d = 0 \mid Y_1 = y_1)$$

is the negative of denominator, thus $P(Z \mid Y_d = 1) = P(Z \mid Y_d = 0)$ contradicts that $Z \not\perp\!\!\!\perp Y_d$, which completes the proof. □

# H   Proof of Theorem C.7

First we bound the sample complexity of the backward phase of IAMB, as shown in Algorithm 5.

**Lemma H.1.** *Let $\widehat{m}_{jk}$ be the output of Algorithm 5 with $\kappa \leq \Delta_I/2$ and the plug-in mutual information estimator* (10). *Then*

$$P(\widehat{m}_{jk} = m_{jk} = \mathrm{pa}(k)) \geq 1 - \frac{|A_j| \delta_{|A_j|}^2}{\kappa^2}$$

*where $\delta_{A_j}^2$ follows the definition in* (9) *with $|S|$ replaced by $|A_j|$.*

*Proof.* Following the same argument in (10), we have

$$\sup_{\ell \in A_j} P\left( \left| \widehat{I}(X_k; X_\ell \mid A_j \setminus X_\ell) - I(X_k; X_\ell \mid A_j \setminus X_\ell) \right| \geq t \right) \leq \frac{\delta_{|A_j|}^2}{t^2}$$

Note that for nodes $X_\ell$ not in $m_{jk}$,

$$I(X_k; X_\ell \mid A_j \setminus X_\ell) = I(X_k; X_\ell \mid m_{jk}) = 0$$

Therefore with probability at least $1 - |A_j| \delta_{|A_j|}^2/t^2$, we have for all $\ell \in A_j$

$$|\widehat{I}(X_k; X_\ell \mid A_j \setminus X_\ell) - I(X_k; X_\ell \mid A_j \setminus X_\ell)| < t$$

Therefore, for $\ell \in m_{jk}$ and $\ell' \in A_j \setminus m_{jk}$,

$$\widehat{I}(X_k; X_\ell \mid A_j \setminus X_\ell) > \Delta_I - t \quad \widehat{I}(X_k; X_{\ell'} \mid A_j \setminus X_{\ell'}) < t$$

Let $t = \kappa \leq \Delta_I/2$, we can remove all $\ell \in A_j \setminus m_{jk}$ rather than any one in $m_{jk}$, then the desired Markov boundary is recovered. □

We are now ready to prove Theorem C.7.

*Proof of Theorem C.7.* Then we use Lemma H.1 and take intersection over all nodes

$$P\left(\widehat{m}_{jk} = \mathrm{pa}(k) \ \forall k \in [d]\right) \geq 1 - \frac{d}{\kappa^2} \max_j |A_j| \delta_{|A_j|}^2 \geq 1 - \frac{d^2}{\kappa^2} \delta_d^2$$

where

$$\delta_d^2 \asymp \left(\frac{2^d}{nd}\right)^2 + \frac{d^2}{n}$$

The last inequality is due to $|A_j| \leq |A_{r-1}| \leq d$. Furthermore, use the estimator (7) for conditional entropy directly on $A_j$ instead of $m_{jk}$,

$$\sup_{k \in V \setminus A_j} P\left(\left|\widehat{H}(X_k \mid A_j) - H(X_k \mid A_j)\right| \geq t\right) \leq \frac{\delta_{|A_j|}^2}{t^2}.$$

Following the proof of main theorem in Appendix E, we can show that without PPS procedure, we can recover layers with

$$P(\widehat{L} = L) \geq 1 - \frac{4dwr}{(\Delta/2 \wedge \omega)^2} \max_j \delta_{|A_j|}^2 \geq 1 - \frac{4dwr}{(\Delta/2 \wedge \omega)^2} \delta_d^2.$$

Thus we have recovery for the whole graph with

$$P\left(\widehat{G} = G\right) = P\left(\widehat{L} = L\right) P\left(\widehat{MB} = \mathrm{pa}(k) \ \forall k \in [d] \mid \widehat{L} = L\right)$$

$$\geq 1 - \left(\frac{d^2}{\kappa^2} + \frac{4dwr}{(\Delta/2 \wedge \omega)^2}\right) \delta_d^2 \gtrsim 1 - \frac{dwr}{(\min(\kappa, \Delta/2, \omega))^2} \delta_d^2$$

Plug in the upper bound of $\delta_d^2$, require for $P(\widehat{G} = G) > 1 - \epsilon$, we have desired sample complexity. $\square$

# I Experiment details

We describe the details of experiments conducted in this appendix.

## I.1 Experiment settings

For graph types, we generate

- *Poly-Tree (Tree).* Uniformly random tree by generating a Prüfer sequence with random direction assigned for each edge.
- *Erdős Rényi (ER).* Graphs whose edges are selected from all possible $\binom{d}{2}$ edges independently with specified expected number of edges.
- *Scale-free networks (SF).* Graphs simulated according to the Barabasi-Albert model.

For models, we consider the dependency between parents and children. We control the constant conditional entropy $H(X_k \mid \mathrm{pa}(k))$ for all $k = 1, 2, \ldots, d$ to satisfy the Condition (C3), which implies (C1). We generate data from following models

- mod model (MOD): $X_k = (S_k \mod 2)^{Z_k} \times (1 - (S_k \mod 2))^{1-Z_k}$ where $S_k = \sum_{\ell \in \mathrm{pa}(k)} X_\ell$ with $Z_k \sim \mathrm{Ber}(p)$
- additive model (ADD): $X_k = \sum_{\ell \in \mathrm{pa}(k)} X_\ell + Z_k$ with $Z_k \sim \mathrm{Ber}(p)$

Total number of replications is $N = 30$. For each of them, we generated random datasets with sample size $n \in \{1000, 2000, 3000, 4000\}$ for graphs with $d \in \{10, 20, 30, 40, 50\}$ nodes and $p = 0.2$.

## I.2 Implementation and baselines

We implement our algorithm with entropy estimator proposed in Wu and Yang [47], which is available at `https://github.com/Albuso0/entropy`. We treat joint entropy as a multivariate discrete variable to estimate. We fix $\kappa = 0.005$ and $\omega = 0.001$, in particular, we do no hyperparameter tuning.

We further compare following DAG learning algorithms as baselines:

- PC algorithm is standard structure learning approach and The implementation is available at `https://github.com/bd2kccd/py-causal`. The independence test is chosen as discrete BIC test `dis-bic-test`. Remaining parameters are set as default or recommended in tutorial.

- Greedy equivalence search (GES) is standard baseline for DAG learning. The implementation is available at `https://github.com/bd2kccd/py-causal`. The score is chosen as discrete BIC score `dis-bic-score`. Remaining parameters are set as default or recommended in tutorial.

The experiments were conducted on an internal cluster using an Intel E5-2680v4 2.4GHz CPU with 64 GB memory.