# OpenReview forum: "Efficient Bayesian network structure learning via local Markov boundary search"
_NeurIPS.cc/2021/Conference — NeurIPS 2021 Poster_

### Official Review · Reviewer_aYyX · 2021-07-16

**Rating:** 6
**Confidence:** 3

**Summary:**

The paper studies the problem of learning the Bayesian network structure of $d$ random variables in the fully observational case.

Among the main contributions, the authors propose:

1. A novel identifiability condition for BN structure learning which does not require the typical faithfulness condition. This new condition is based on conditional entropies of the random variables given their ancestors.

2. Motivated by the proposed conditions in Theorem 3.1, authors show Algorithm 1 to learn the BN structure, which is based on estimating the Markov boundary of each node with respect to its ancestor set.

3. Computational and statistical complexities are discussed, and they are polynomial when the maximum Markov boundary size is $O(\log d)$.

4. Some synthetic experiments are provided and the proposed method is compared against the classical PC and GES algorithms.


**Limitations And Societal Impact:**

Authors have discussed some limitations of their methods. Being the work of theoretical nature, I do not foresee any negative societal impact.

**Main Review:**

Here I present my review along the axis of the NeurIPS reviewing guidelines:

**Originality**

- Are the tasks or methods new?
  - The most novel contribution relates to Theorem 3.1, which is the identifiability condition based on conditional entropies. Under this condition, faithfulness is not required which I find appealing.
The proposed algorithm for learning the structure is not quite novel  on its own in the sense that there are existing work that also learn the network in a layer-wise fashion; similarly, as stated by the authors, learning the Markov boundary has also been studied before.

- Is the work a novel combination of well-known techniques?
  - Given the new identifiability condition, I think Algorithm 1 can be regarded as a novel combination of known ideas for learning the BN structure. Moreover, the algorithm requires several estimations of conditional entropies, for which the authors use existing estimation techniques.

- Is it clear how this work differs from previous contributions?
  - Yes. The main contrast to previous work is with regards to the identifiability condition.

- Is related work adequately cited?
  - Yes, although I believe there are some lines of work that have not been commented whatsoever. For instance, learning BN structures from continuous optimization has gained much attention in recent years but they are not been mentioned.


**Quality**

- Is the submission technically sound?
  - Yes, claims are sound.

- Are claims well supported (e.g., by theoretical analysis or experimental results)?
  - Yes, proofs are provided in the appendix. I have checked most of them and could not find major issues.
Synthetic experiments are also provided although I do not find them convincing.

- Are the methods used appropriate?
  - Yes.

- Is this a complete piece of work or work in progress?
  - Complete work.

- Are the authors careful and honest about evaluating both the strengths and weaknesses of their work?
  - Yes. Although some important discussion could be missing. See my questions at the end.


**Clarity**

- Is the submission clearly written?
  - Yes. I was able to follow the contributions without any problems. Can be improved, see last section of my review.

- Is it well organized?
  - Yes.

- Does it adequately inform the reader?
  - Yes.


**Significance**

- Are the results important?
  - I believe the identifiability condition is interesting and important. The algorithm is also compelling; however, I do have some questions that would like to be addressed. See below.

- Are others (researchers or practitioners) likely to use the ideas or build on them?
  - Maybe. There are open questions that some researchers might find interesting to study.

- Does the submission address a difficult task in a better way than previous work?
  - Unclear. The experiments are not convincing on this.

- Does it advance the state of the art in a demonstrable way?
  - No. Relying in the experiments shown, the proposed seems to be at best comparable.

- Does it provide unique data, unique conclusions about existing data, or a unique theoretical or experimental approach?
  - The work is mainly theoretical with a small set of experiments.

# Questions / Comments

1. One of the main arguments of the new identifiability condition is that it applies to general distributions without parametric assumptions. However, the conditions are given in the population case. I believe that the set of unfaithful distributions has Lebesgue measure 0 in the population case. What would be the advantage of the new condition against faithfulness if it is given in the population setting?

2. In Lines 92-93 it is stated that the proposed approach highlights the role of fundamental assumptions obscured in the linear case. Can authors elaborate on this? I could not find a proper discussion of this claim in later sections of the paper.

3. It is not until page 7, line 261, where one notices that there is a "sparsity" assumption of $M \leq \log d$. I believe this is an important assumption that should be stated in the abstract and contributions section. If this assumption does not hold, the sample and time complexity are exponential.

4. Right before line 835, the error bound is confusing to me. Does it decay?

5. The experiments are unconvincing. First, the method seems to work worse than PC and GES in the low sample complexity regime, which is an important drawback. Also, as the number of samples increases, the proposed method is at best comparable. Both reasons undermine the usability of the method in practice. Second, given the sample and time complexity results, it would be interesting to see the log-dependence/phase-transitions plots. Finally, it would help to compare against methods that provide guarantees such as that of Ghoshal & Honorio.

Minor typos/comments:

Line 135: it would perhaps help to define $\lceil a \rceil^+ \equiv \max (a,0)$, and use $\lceil L(X_k) - 2 \rceil^+$ to avoid $L(X_k) - 2$ being negative.

Line 191: it says "we use apply..."

Line 254: it says "...any any.."

---
I thank the authors for their response. I feel more positive about the paper and will increase my score to 6. I think that the paper could benefit by adding some points of this discussion.




**Time Spent Reviewing:**

6

---

> ### Author Response · Authors · 2021-08-10
> **Response to Reviewer aYyX**
>
> We thank the reviewer for the helpful comments. Please find our response below:
>
> **Compared to faithfulness**: This is an insightful question, and we acknowledge that studying a direct comparison between our assumptions and faithfulness is an interesting question. Here we can make a few comments.
>
> At the population level, we agree that our condition (C2) is superficially similar to faithfulness, however, a closer look reveals significant differences. Please see L141-147 for a discussion: Violating (C2) requires _multiple_ path cancellations to conspire together simultaneously (descendant vs. whole layer of ancestors), whereas unfaithfulness occurs when there happens to be just one such cancellation. Moreover, (C2) only applies to a small set of ancestral sets, whereas faithfulness applies to all possible d-separating sets. Furthermore, in Appendix G, we show that (C2) _always_ holds for polyforests.
>
> On finite samples, of course we actually need _strong_ faithfulness, i.e. the DAG is bounded away from being unfaithful. Therefore, although unfaithfulness space is measure zero, the space of practically learnable DAGs can be very small [1,2]. This has been widely discussed and criticized as a strong assumption [1,2]. By contrast, our corresponding finite sample assumption is that $\Delta>0$ and $\eta>0$, which only require a much smaller, restricted set of information measures (note the inner min over $k\in V \setminus A_{j+1}$) are positive. Finally, it is easy to construct simple unfaithful examples where PC/GES may fail; please see below.
>
> **Role of fundamental assumption**:  The linear case [3], makes very strong use of linearity, independence, and additivity. Briefly, the relation $\text{var}(X_j) = \text{var}(\beta_j^\top X) + \text{var}(\epsilon_j)$ is crucial in these arguments (perhaps implicitly). Furthermore, the linear case cleverly exploits the covariance structure of linear models, which does not apply in our setting. By contrast, our approach reveals that it is enough to exploit generic *uncertainty* accumulation as the system evolves (i.e. according to its layer decomposition / time ordering), without making any linearity (or even second moment) assumptions.
>
> At an intuitive level, this can be summarized as follows: Every distribution with $\mathbb{E} X^\epsilon < \infty$ for some $\epsilon>0$ has well-defined information measures, whereas not all distributions have a linear covariance structure; indeed general distributions need not even have second moments. Our proofs rely only on the former, whereas existing works carefully exploit the latter.
>
> **Sparsity assumption and Error bound**: The logarithmic dependence on $d$ in $M$ is used to simplify the final result. We note that $\Omega(2^M)$ samples are in fact necessary in light of known results from nonparametric statistics, so this is not a limitation per se but a fact of life when dealing with nonparametric models. Nonetheless, we agree this should be stated or at least hinted in early sections, and will correct this in the final version.
>
> The $\delta$ is the MSE of entropy estimation, which is assumed to go to zero faster than the factor before it. Thanks for pointing these issues out!
>
> **Experiments**: We stress that the current experiments are used for simple illustration thus not well-optimized compared to existing algorithms or fully exploited the main condition either (see L306-308). We have made no attempts whatsoever to optimize the algorithm or entropy estimation, tune parameters, or devise clever models. By contrast, the PC and GES implementations are highly optimized to be stable and fast. We speculate that the comparatively poor performance comes from the hardness of entropy estimation and the fact that our synthetic models are all in fact faithful. As another reviewer has pointed out, a more convincing demonstrating of our algorithm is on **unfaithful** models, which we are happy to include. On such models (which are extremely simple to derive, see below), both PC and GES will fail while our algorithm succeeds. Furthermore, these simple examples can be embedded into any model. We are happy to talk more about this in detail.
>
> **Example**: We conclude with a simple example where PC/GES fail but our algorithm succeeds (both in theory and in practice). Consider a notorious three-node DAG: $Z\to Y$, $Z\to X\to Y$, where $Z$ is a common cause of $X$ and $Y$, and the effect $Z\to Y$ is cancelled by the path $Z\to X\to Y$. Then independence between $Z$ and $Y$ does not imply the d-separation between them. Now add one more node $W$ with an edge $W\to X$. This Bayesian Network is still unfaithful due to the independence between $Z$ and $Y$, but can easily be made to satisfy our identifiability condition. To see this, in the first layer, $W$ is used to mask $(X,Y)$ (note that $Z$ is unable to, by independence). Then in the second layer, $X$ will mask $Y$. We have run experiments to show PC/GES will have SHD around 3 (i.e. very bad for this small model) but our algorithm will go to zero as expected.
>
> **Time complexity**: We include a quick experiment to show the time complexity of our algorithm. We consider one particular setting in the experiment, which is ER graph with MOD model, sample size $n=3000$. Apply our algorithm on dimension $d=5,10,20,30,40$ and record the average running time (in seconds) over 10 replications. Results are summarized in the following table. More experiments on time complexity will be appended in the final version.
>
> | $d$ | 5  | 10 | 20 | 30 | 40 |
> | --- | --- | --- | --- | --- | --- |
> | runtime |0.106|  0.624|  2.79|  7.51| 13.8|
>
> **Literature in continuous optimization**: We thank the reviewer for mentioning this line of research. Although we are mainly focused on comparing methods with finite sample guarantees while continuous optimization has---to the best of our knowledge---no finite sample guarantees, they are definitely interesting and we are happy to add these citations.
>
> Finally, we appreciate the helpful  advice on the writing.
>
> **Reference**:
>
> [1] Robins, James M., et al.  Uniform Consistency in Causal Inference. https://www.jstor.org/stable/30042062
>
> [2]  Uhler, Caroline, et al. Geometry of the faithfulness assumption in causal inference. https://arxiv.org/abs/1207.0547
>
> [3] W. Chen, M. Drton, and Y. S. Wang. On causal discovery with an equal-variance assumption. https://arxiv.org/abs/1807.03419

---

### Official Review · Reviewer_kVE1 · 2021-07-16

**Rating:** 6
**Confidence:** 4

**Summary:**

It analyzes the complexity of learning directed acyclic graphical (DAG) models from observational data.  It proposes an algorithm that uses a local Markov boundary search to learn an identifiable DAG G from samples. They prove an identifiability result on DAG learning. Roughly speaking, this condition requires that the entropy conditioned on an ancestral set of each node in G is dominated by one of its ancestors (Condition 1). They, also derive sample complexity of their algorithm and further illustrate its performance in a simulation study.

**Limitations And Societal Impact:**

Yes. Please see my comments.

**Main Review:**

It studies an interesting and important problem in causal structure learning.

The paper is well-written but more explanation about the role of Masked set in Algorithm 1 would improve its readability.

The theoretical results especially the sample complexity results are quite valuable and could be beneficial
 to the researchers in this field.
However, my main concern is about the conditions under which the results are presented and also the performance of the proposed algorithm.

Although, it is discussed that C1 is a generalization of equal entropy condition but having C1 or PPS condition heavily depends on the underlying structural equation model (SEM) that generates the data. Moreover, how verifiable are these conditions given observational data?
What type of systems would satisfy conditions 1 and 2? For instance, linear with non-Gaussian systems etc.

Although the authors claim that their approach does not directly assume faithfulness but having conditions 1 and 3 somewhat encourages similar phenomena (I do not suggest that these conditions and faithfulness are equivalent). It would be educational to see an example that violates faithfulness but satisfies the conditions 1 and 3. Is MOD model in the experimental section an example?

A brief discussion about the result of proposition 4.1. is required. For instance, when does the right-hand-side of the equation in this proposition is positive? It seems that often (e.g., for Erdos-Renyi graphs) this bound is trivial, i.e., it is negative. Clearly, it depends on the parameters.

Given the experimental results, the proposed algorithm performs often as good as PC and GES in large sample sizes but for smaller sample size (1000) it often performs poorly. How does Algorithm 1 compare to the others in terms of run-time and computational complexity? Also, it is important to see (e.g., empirically) how tight is the proposed sample complexity in Theorem 5.1.


**Time Spent Reviewing:**

4

---

> ### Author Response · Authors · 2021-08-10
> **Response to Reviewer kVE1**
>
> We thank the reviewer for the helpful comments. Please find our response below:
>
> **Condition 1 and 2**: Condition 2 is originated from [1,2], where we generalize the variance to entropy. (C1) of Condition 1 further covers Condition 2 through Theorem B.6. Thus any system satisfying the intuition that the entropy (variance) of the variables is increasing as the times ordering evolves would meet our conditions, which definitely covers linear non-Gaussian system etc. The additive noise model and the MOD model considered in the experiment also serve as examples. We agree that our condition (C2) is similar in some sense with faithfulness, but much weaker. See L141-147 for a discussion; violating (C2) requires _multiple_ path cancellations to conspire together simultaneously (descendant vs. whole layer of ancestors), whereas unfaithfulness occurs when there happens to be just one such cancellation. Furthermore, in Appendix G, we show that (C2) _always_ holds for polyforests.
>
> **Unfaithful example**: As the reviewer brought up, we are also happy to share a simple unfaithful example for PC/GES to fail but still satisfies our condition. Consider a notorious three-node DAG: $Z\to Y$, $Z\to X\to Y$, where $Z$ is a common cause of $X$ and $Y$, and the effect $Z\to Y$ is cancelled by the path $Z\to X\to Y$. Then independence between $Z$ and $Y$ does not imply the d-separation between them. Now add one more node $W$ with an edge $W\to X$. This Bayesian Network is still unfaithful due to the independence between $Z$ and $Y$, but can easily be made to satisfy our identifiability condition. To see this, in the first layer, $W$ is used to mask $(X,Y)$ (note that $Z$ is unable to, by independence). Then in the second layer, $X$ will mask $Y$. We have run experiments to show PC/GES will have SHD around 3 (i.e. very bad for this small model) but our algorithm will go to zero as expected.
>
> This simple example can be embedded into any DAG satisfying our identifiability condition by treating some sink node as $W$. And we may conjecture that, when unfaithfulness happens in the form of path-cancellation, if we have some other nodes within the “ancestral” layer, it is able to help identify the DAG as long as the  “uncertainty” relation is satisfied (C1).
>
> **Tightness of sample complexity**: Since our setting is fully nonparametric, exponential rates in the dimensions are to be expected. Indeed, we can show there is a lower bound $\Omega(2^M)$ on the sample complexity by appealing to known results from nonparametric statistics.
>
> **Proposition 4.1**: Basically, we need the estimation error $\delta$ here to be sufficiently small and go to zero faster than the factor before it. Thanks for pointing out and we will stress it in the final version.
>
> Finally, we appreciate the helpful  advice on the writing.
>
> **Reference**:
>
> [1] W. Chen, M. Drton, and Y. S. Wang. On causal discovery with an equal-variance assumption. https://arxiv.org/abs/1807.03419
>
> [2] M.  Gao,  Y.  Ding,  and  B.  Aragam.   A  polynomial-time  algorithm  for  learning  nonparametric  causal graphs. https://arxiv.org/abs/2006.11970

---

### Official Review · Reviewer_WuUC · 2021-07-23

**Rating:** 5
**Confidence:** 4

**Summary:**

The paper proposes a polynomial time algorithm for learning very sparse DAGs (constant maximum Markov blanket size) from observational data under a new identifiability condition that generalizes some of the existing identifiability condition (especially for the linear Gaussian case). The authors also derive sample complexity of their algorithm which has logarithmic dependence on the number of variables but unfortunately exponential dependence on the maximum MB size. The strengths of the paper are that the authors make no distributional (e.g. Gaussian) or functional (e.g. linear additive model) assumptions for deriving their identifiability condition and the associated algorithm. The algorithm learns a DAG layer-wise and there is a novel masking strategy due to which they identify the layers without requiring that all the nodes have the same conditional entropy. The paper advances the identifiability theory of general Bayesian networks and proposes a novel algorithm. However, even though the method is general and the assumptions are purely information-theoretic, it is not clear if the identifiability condition is useful beyond the linear Gaussian case — non-linear additive noise models and linear non-Gaussian (LinGAM) models are identifiable without any stringent conditions. Furthermore, the exponential dependence of the sample and computational complexity on the MB size is problematic, the experiments are not very strong, and lastly the presentation is too dense and unintuitive.

**Limitations And Societal Impact:**

Yes.

**Main Review:**

## Strengths

1. General algorithm and no distributional assumptions. The authors do not even assume additive models. The conditions are fully information-theoretic.
2. The core idea is novel: that once you have identified 0 to j layers, in order to identify the (j+1)-th layer we don't need every node after the (j+1)-th layer to have larger conditional entropy than the nodes in the (j+1)-th layer (whose parents have been identified and hence the conditional entropy is the entropy of the exogenous/noise variable in an additive model). As long as there is one node in the (j+1)-th layer for which this condition is satisfied, we can mask out it's descendants.

## Weakness

- Why not just use the PC algorithm to first estimate the skeleton ? Then, for a node $X_i$, the intersection of its neighbors (in the skeleton) with the layers identified so far should give its parents.
- The current algorithm's computational and sample complexity is exponential in the maximum size of a node's ancestral Markov blanket (M). Whereas the PC algorithm's computational complexity is exponential in the tree-width and sample complexity is polynomial in the maximum neighborhood size ("Estimating High-Dimensional Directed Acyclic Graphs
with the PC-Algorithm", Kalisch and Buhlmann). In addition the PC algorithm doesn't need Condition 3. The PC algorithm doesn't make any distributional or functional assumptions and only requires faithfulness to identify the skeleton.
- How does the identifiability condition fit with existing identifiability results ? We know it generalizes the identifiability condition of Ghoshal and Honorio 2018 (GH2018) and equal variance condition for linear Gaussian case. But we know that linear non-Gaussian (LinGAM) models are identifiable and non-linear additive noise models are identifiable. Polynomial time algorithms exist for both these cases (ICA based method for LinGAM and RESIT for non-linear ANM). For LinGAM and non-linear ANM models the proposed identifiability condition is very strong and is only sufficient but not necessary. In spite of making no distributional and parametric assumptions, looks like the only case where it is useful is linear Gaussian case.
- The presentation can be significantly improved. Notation is too dense and confusing
    - Rather than saying 'the "$X_i$" for "$X_k$"' introduce a term for their relationship and use that.
    - Say what "testing and masking" means
    - Line 176: $d_6$ should be $h_6$.
    - Line 177: "$h_3 < h_{[4:6]}$ and $h_5 < h_6$  imply condition 2 doesn't hold here". Why ? Condition 2 is about conditional entropy of a node given parents, where $h_3, ..., h_6$ are marginal entropies.
- Somehow identifiability condition doesn't significantly relax the identifiability condition of GH2018 for linear SEMs. For a fully connected DAG (i.e. whose skeleton is a complete graph) it looks like the two identifiability condition are the same. The identifiability condition is significantly weaker than GH2018 only for very sparse graphs.
- Experiments: running time comparison with PC and GES (surprisingly GES does well even in spite of returning a local optima).
- Experiments: no experiments when identifiability condition is not satisfied. So it is not clear if how often the condition is satisfied for sparse and dense graphs.
- Experiments: Even when the identifiability condition is satisfied the algorithm doesn't significantly out-perform PC or GES. Why ? This is in spite of the fact that the proposed algorithm specifically exploits the condition. Does the SHD go to zero as the number of samples is further increased ?

**Time Spent Reviewing:**

4

---

> ### Author Response · Authors · 2021-08-10
> **Response to Reviewer WuUC**
>
> We thank the reviewer for the helpful comments and for acknowledging the novelty of our approach. Please find responses to your questions below.
>
> **PC algorithm and Faithfulness**: Though simple and intuitive, the PC algorithm requires assuming (strong) faithfulness, which has been widely discussed and criticized as a strong assumption [1,2]. On finite samples, even under the Gaussian linear setting in [3], PC actually needs _strong_ faithfulness, i.e. all partial correlations are bounded away from 0. In fact, it is easy to construct a simple unfaithful example where PC/GES may fail; please see below.
>
> **Compared with existing results**: It is a pleasure to see the generalization of existing work (GH2018) is recognized and we re-emphasize again by pointing to Theorem B.6. We believe Condition 1 significantly generalizes previous work in literature [4,5], and we provide examples in Appendix B.3 (in fact, the unfaithful example below can be made as another one to show our significance). Apart from that, our condition works for very general models including LinGAM and nonlinear ANM mentioned by the reviewer. Indeed, it is only sufficient but not necessary. Although it is true that LinGAM and RESIT are poly-time in the infinite-date (i.e population) setting, these algorithms do not have _finite-sample_ poly-time guarantees. Moreover, our condition does not require independent or additive noise in the ANM, which is the basic property RESIT relies on. Thus, our results also apply to models which RESIT cannot identify.
>
> **Experiment**: We stress that the current experiments are used for simple illustration thus not well-optimized compared to existing algorithms or fully exploited the main condition either (see L306-308). We have made no attempts whatsoever to optimize the algorithm or entropy estimation, tune parameters, or devise clever models. By contrast, the PC and GES implementations are highly optimized to be stable and fast. We speculate that the comparatively poor performance comes from the hardness of entropy estimation and the fact that our synthetic models are all in fact faithful. As another reviewer has pointed out, a more convincing demonstrating of our algorithm is on **unfaithful** models, which we are happy to include. On such models (which are extremely simple to derive, see below), both PC and GES will fail while our algorithm succeeds. Furthermore, these simple examples can be embedded into any model. We are happy to talk more about this in detail.
>
> **Example**: We conclude with a simple example where PC/GES fail but our algorithm succeeds (both in theory and in practice). Consider a notorious three-node DAG: $Z\to Y$, $Z\to X\to Y$, where $Z$ is a common cause of $X$ and $Y$, and the effect $Z\to Y$ is cancelled by the path $Z\to X\to Y$. Then independence between $Z$ and $Y$ does not imply the d-separation between them. Now add one more node $W$ with an edge $W\to X$. This Bayesian Network is still unfaithful due to the independence between $Z$ and $Y$, but can easily be made to satisfy our identifiability condition. To see this, in the first layer, $W$ is used to mask $(X,Y)$ (note that $Z$ is unable to, by independence). Then in the second layer, $X$ will mask $Y$. We have run experiments to show PC/GES will have SHD around 3 (i.e. very bad for this small model) but our algorithm will go to zero as expected.
>
> **Line 177**: If Condition 2 is satisfied, $h_4=h_6=H(X_3|X_2,X_5) < h_3$. The last inequality is by $h_3 = H(X_3|X_2,X_5) + I(X_3;(X_2,X_5))$  and positivity of mutual information.
>
> Finally, we appreciate the helpful  advice on the writing.
>
> **Reference**:
>
> [1] Robins, James M., et al.  Uniform Consistency in Causal Inference. https://www.jstor.org/stable/30042062
>
> [2]  Uhler, Caroline, et al. Geometry of the faithfulness assumption in causal inference. https://arxiv.org/abs/1207.0547
>
> [3] M Kalisch, P Bühlman. Estimating High-Dimensional Directed Acyclic Graphs with the PC-Algorithm. https://arxiv.org/abs/math/0510436
>
> [4] W. Chen, M. Drton, and Y. S. Wang. On causal discovery with an equal-variance assumption. https://arxiv.org/abs/1807.03419
>
> [5] M.  Gao,  Y.  Ding,  and  B.  Aragam.   A  polynomial-time  algorithm  for  learning  nonparametric  causal graphs. https://arxiv.org/abs/2006.11970

---

### Official Review · Reviewer_Wj5d · 2021-07-26

**Rating:** 7
**Confidence:** 2

**Summary:**

The authors present several novel results regarding learning of discrete graphical models.  They introduce a conditional entropy-based identifiability condition that allows one to decompose an unknown directed acyclic graph (DAG) structure into different layers.  They introduce another entropy based measure to allow them to efficiently locate the markov boundary of nodes in each layer without backtracking or pruning.  These two results are combined to develop a novel algorithm for learning a DAG from sample data.  The authors demonstrate the correctness of the proposed algorithm by empirically verifying that the procedure recovers the expected graphs when the required assumptions are met.  Finally, they introduce a sample complexity result for how quickly one can recover the minimial IMAP G, as a function of the number of nodes and number of layers in the DAG.

These contributions seem significant and, to the best of my knowledge, are novel.

**Main Review:**

My main question/criticism for the authors is how likely are conditions 1 and 3 to be met in practice and what happens to the proposed procedure when these conditions are not satisfied? Do you empirically get sensible results, or results on par with other learning algos, when running on graphs that don't satisfy this criteria?  If condition 3 is relaxed, do you simply lose the sample complexity guarantee?

The proposed algorithm seems to perform worse than PC and GES for small sample size and larger number of nodes. Can you provide any color as to why this is the case?

**Time Spent Reviewing:**

1

---

> ### Author Response · Authors · 2021-08-10
> **Response to Reviewer Wj5d**
>
> We thank the reviewer for the helpful comments and overall positive review. Please find our response to your questions below.
>
> **Conditions 1 and 3**: The rationality and robustness of a proposed condition is always a fair question to ask, and we thank the reviewer for bringing this up. Indeed, we have gone to great lengths in Appendix B to discuss these assumptions, including numerous explicit examples and generalizations in 6 subsections. For example, we believe Condition 1 significantly generalizes (Appendix B.3) previous work in literature [1,2], and we provide examples (both satisfied and violated) and further extensions in Appendix B. For Condition 3, poly-forest is a large class of DAG provably satisfying it (Sec 5.3, Theorem 5.2). Even when the DAG is not a polyforest, we have included additional examples in Appendix B.4 under which the PPS condition holds. Empirically, although ER/SF graphs do not necessarily satisfy Condition 3, they seem to work well often. Hopefully this provides convincing evidence that our assumptions are both interesting and robust to generalization.
>
> **Experiments**: We stress that the current experiments are used for simple illustration thus not well-optimized compared to existing algorithms or fully exploited the main condition either (see L306-308). We speculate that the comparatively poor performance comes from the hardness of entropy estimation and the fact that our synthetic models are all in fact faithful. For completeness, below we illustrate a simple example where both PC and GES will fail while our algorithm succeeds. Furthermore, these simple examples can be embedded into any model. We are happy to talk more about this in detail.
>
> **Example**: We conclude with a simple example where PC/GES fail but our algorithm succeeds (both in theory and in practice). Consider a notorious three-node DAG: $Z\to Y$, $Z\to X\to Y$, where $Z$ is a common cause of $X$ and $Y$, and the effect $Z\to Y$ is cancelled by the path $Z\to X\to Y$. Then independence between $Z$ and $Y$ does not imply the d-separation between them. Now add one more node $W$ with an edge $W\to X$. This Bayesian Network is still unfaithful due to the independence between $Z$ and $Y$, but can easily be made to satisfy our identifiability condition. To see this, in the first layer, $W$ is used to mask $(X,Y)$ (note that $Z$ is unable to, by independence). Then in the second layer, $X$ will mask $Y$. We have run experiments to show PC/GES will have SHD around 3 (i.e. very bad for this small model) but our algorithm will go to zero as expected.
>
> **Reference**:
>
> [1] W. Chen, M. Drton, and Y. S. Wang. On causal discovery with an equal-variance assumption. https://arxiv.org/abs/1807.03419
>
> [2] M.  Gao,  Y.  Ding,  and  B.  Aragam.   A  polynomial-time  algorithm  for  learning  nonparametric  causal graphs. https://arxiv.org/abs/2006.11970

---

### Official Review · Reviewer_YmV1 · 2021-08-01

**Rating:** 5
**Confidence:** 4

**Summary:**

Learning Bayesian networks is a notoriously difficult task, considering the fact that the search space for all possible graphs scales exponentially with the potential arcs. In the paper, authors propose a heuristic search algorithm which reduces the time complexity of well-known exact algorithms such as PC. PC is a constraint-based algorithm which recovers Bayesian networks based on the conditional independencies entailed by the nodes of the graphs. Whereas the methodology proposed in the paper makes an additional information theoretical assumption for the structure of the graph, i.e. the conditions stated in the paper, which effectively reduces the search space of valid graphs for the given nodes and enables us to make use of a greedy search algorithm.

**Limitations And Societal Impact:**

I haven’t seen a part where the limitations and the social impact of the work are mentioned.

**Main Review:**

Paper is easy to read and very well written in general. Assumptions, theorems, derivations, and algorithms are stated in full clarity. Authors also demonstrate their algorithm on a toy example, which is very helpful for understanding the main principles of the algorithm. I think such a paper will be of interest to NeurIPS community.

The main contribution of the paper is the information theoretical conditions that are proposed to narrow down the overall search space of graphs. In the first phase of the proposed methodology, a topological ordering of the nodes, i.e. the layers of the underlying graph, is found using these conditions. Once the topological ordering is found, the rest of the graph search becomes straightforward, Namely, the edges between the layers of the graph are searched using the conditional independence properties.

However, any topological ordering enforced on a graph would yield a search procedure with the same efficiency as the methodology in the paper. Indeed, one could change the conditional H with any functional F, say negative H, and would still come up with a unique DAG, because there is nothing specific to H in the proposed methodology. So, I believe the selection of H needs further justification, e.g. how does the selected criteria relate to the actual topological ordering of the nodes? What are the advantages of choosing H for narrowing down the search space rather than a more general functional F?

Additionally, it is not perfectly clear to me that why the size of largest markov boundary can be assumed less than logarithm of the number of nodes. This is a critical assumption for calculation of the polynomial time complexity stated in the paper, and I think an addittional justification for this would be helpful, too.

**Time Spent Reviewing:**

10

---

> ### Author Response · Authors · 2021-08-10
> **Response to Reviewer YmV1**
>
> We thank the reviewer for the helpful comments, especially the positive comments regarding the writing and clarity. Before responding point by point, we would like to emphasize up front that our approach is **not** heuristic, but comes with provable guarantees and a rigorous analysis of every assumption, including possible generalizations (see Appendix B). We will be happy to discuss our theoretical results in more detail during the discussion phase.
>
> **Choice of $H$**: Indeed, the entropy function $H$ is not necessarily the only choice for this ordering search based algorithm. Functionals like variance [1], or other more complicated functionals [2,3] can be applied as well. But in order to have this greedily minimization to work, the functional $F$ ought to have a property that $F(X_k|A) \ge 0$ for any nondescendant set $A$. And the equality holds iff $\text{pa}(k) \subseteq A$. Both conditional entropy and variance satisfy this property. Regarding entropy specifically, entropy generalizes nicely to general families of distributions (entropy only requires _some_, possibly fractional, moment to exist, i.e. $\mathbb{E} X^\epsilon < \infty$ for some $\epsilon>0$) and has a known optimal estimator. For other more complicated functionals, optimal estimation may not be known.
>
> **Size of Markov boundary**: We thank the reviewer for commenting on this important point, and ultimately, the logarithmic dependence results from not making parametric assumptions on the model family. As such, our setting is fully nonparametric, and exponential rates in the dimensions are to be expected. Indeed, we can show there is a lower bound $\Omega(2^M)$ on the sample complexity by appealing to known results from nonparametric statistics. Under stronger parametric assumptions, these exponential rates can likely be avoided.
>
> **Reference**:
>
> [1] M.  Gao,  Y.  Ding,  and  B.  Aragam.   A  polynomial-time  algorithm  for  learning  nonparametric  causal graphs. https://arxiv.org/abs/2006.11970
>
> [2] G. Park and H. Park.  Identifiability of generalized hypergeometric distribution (ghd) directed acyclic graphical  models. https://arxiv.org/abs/1805.02848
>
> [3] G. Park and G. Raskutti.   Learning quadratic variance function (qvf) dag models via overdispersion scoring (ods).  https://arxiv.org/abs/1704.08783

---

### Decision · Program_Chairs · 2021-09-28

**Decision:**

Accept (Poster)

**Comment:**

The general question of identifying conditions under which Bayesian networks can be learned by efficient algorithms is an important direction that deserves further investigation. The present paper identifies a new condition that roughly amounts to entropies increasing in the layers of the network. Under this condition it is fairly straightforward to recover the network in a greedy fashion layer by layer. The condition is strictly weaker than the well-known faithfulness condition.

The reviewers would have liked to see additional discussion around the conditions for model recovery, including examples and so forth. Along these lines, verifying that the conditions are satisfied in some real-world data would massively strengthen the paper. As it stands, the reviewers opinions remained mixed on the significance of the results and the usefulness of the proposed condition.

**Consistency Experiment:**

NeurIPS has a long history of experimentation. In 2014, NeurIPS ran an experiment in which 10% of submissions were reviewed by two independent committees to quantify the randomness in the review process. This year, we repeated a variant of this experiment to see how the quality of the review process has changed over time.  This paper was part of the experiment and was therefore assigned to two committees (consisting of reviewers, an Area Chair, and a Senior Area Chair) that reached independent decisions.  If both committees made the same recommendation, this recommendation was followed. If a single committee recommended acceptance, the paper was accepted (with the exception of a few cases in which the other committee identified what we considered a fatal flaw, e.g., an error in a key result).

This copy’s committee reached the following decision: **Reject**

The other committee assigned to the paper recommended **Accept (Poster)**.  You can find the other set of reviews, along with any follow up discussion with the authors here:
https://openreview.net/forum?id=fWLDGNIOhYU